# ELPO: Ensemble Learning Based Prompt Optimization for Large Language Models

## Abstract

The remarkable performance of Large Language Models (LLMs) highly relies on crafted prompts. However, manual prompt engineering is a laborious process, creating a core bottleneck for practical application of LLMs. This phenomenon has led to the emergence of a new research area known as Automatic Prompt Optimization (APO), which develops rapidly in recent years. Existing APO methods such as those based on evolutionary algorithms or trial-and-error approaches realize an efficient and accurate prompt optimization to some extent. However, those researches focus on a single model or algorithm for the generation strategy and optimization process, which limits their performance when handling complex tasks. To address this, we propose a novel framework called **E**nsemble **L**earning based **P**rompt **O**ptimization (ELPO) to achieve more accurate and robust results. Motivated by the idea of ensemble learning, ELPO conducts voting mechanism and introduces shared generation strategies along with different search methods for searching superior prompts. Moreover, ELPO creatively presents more efficient algorithms for the prompt generation and search process. Experimental results demonstrate that ELPO outperforms state-of-the-art prompt optimization methods across different tasks, e.g., improving F1 score by 7.6 on ArSarcasm dataset.

## 1 Introduction

Over the past few years, Large Language Models (LLMs) have emerged not merely as incremental improvements in natural language processing (NLP), but as transformative agents redefining the relationship between humans and intelligent systems. Flagship families including GPT (Radford et al., 2018; 2019; Brown et al., 2020; Achiam et al., 2023), LLaMA (Touvron et al., 2023a;b), and PaLM (Anil et al., 2023) are trained on web-scale corpora and display emergent capabilities that are unforeseen in smaller-scale predecessors. Among these, in-context learning (Brown et al., 2020) exemplifies a paradigm shift: models without any fine-tuning can tackle sentiment analysis, text classification, code generation, logical reasoning, and other diverse tasks by following natural language instructions, also known as "prompts".

This ability has fueled visions of a "general-purpose linguistic interface" where machine behavior is shaped as effortlessly as conversing with a colleague. Yet, this promise comes with a sharp problem: LLMs are strikingly sensitive to small changes in prompts (Jiang et al., 2020; Zhao et al., 2021; Lu et al., 2022). Synonym substitutions, minor structural tweaks or rephrased instructions may lead to outputs drastically different from what human intuition expects (Webson & Pavlick, 2022). Such fragility has propelled prompt engineering, the art and science of designing prompts for high-quality outputs into the spotlight (Liu et al., 2023; Reynolds & McDonell, 2021). But for many users, particularly non-experts, crafting effective prompts is an opaque, trial-and-error process, hindered by the vast, unstructured search space of possible natural language instructions (Jiang et al., 2022).

To ease this burden, the field has turned toward Automatic Prompt Optimization (APO) (Zhou et al., 2023). APO automates the prompt design process by creating candidate instructions and identifying the optimal ones through performance evaluation. Strategies ranging from feedback-driven refinement, evolutionary algorithms, to trajectory-based exploration have shown encouraging results. However, they have surfaced some new, fundamental difficulties. First, relying on a single optimization algorithm risks fragility: in light of the "No Free Lunch" theorem for optimization (Wolpert & Macready, 2002), no one strategy can consistently capture every subtlety across tasks.

Second, most existing systems treat the candidate pool as a flat, unstructured set, leading to wasted computation on unpromising variants, thereby diminishing efficiency. These bottlenecks leave APO methods struggling to fulfil the promise of truly adaptive, scalable prompt engineering.

Motivated by both the remarkable potential of LLMs and the instability of prompt-based interaction, we propose a novel framework for APO called **E**nsemble **L**earning based **P**rompt **O**ptimization (ELPO) which combines multiple generation and search algorithms to derive accurate and robust results. As for the prompt generation, three strategies are applied to maintain the diversity and quality of candidate prompts. It is expensive to evaluate each candidate prompt on the entire training dataset (Prasad et al., 2023), so well-designed search methods for minimizing the queries for employing LLMs are also necessary. With respect to prompt search, to the best of our knowledge, we are the first to combine Bayesian search (Jones et al., 1998; Brochu et al., 2010; Snoek et al., 2012) and Multi-Armed Bandit (MAB) (Audibert & Bubeck, 2010; Lattimore & Szepesvári, 2020), applying it to APO for improving search efficiency substantially. Inspired by the idea of ensemble learning (Zhou, 2012), a robust result is chosen by applying multiple generation and search methods along with ensemble voting.

In summary, this paper makes the following main contributions.

(1) As for generation, we creatively propose Hard-Case Tracking which focuses on recurrent error samples and analyzes them in conjunction with failed prompts, employing large language models to generate more robust and generalizable prompts. Moreover, we combine it with other two strategies simultaneously when generating new prompts.

(2) In terms of search efficiency, we propose a novel search algorithm in APO based on Bayesian search. It reflects prompts into high-dimensional spaces to increase search efficiency as a result of evaluation on only part of the prompts.

(3) In terms of robustness and generalization, we use an ensemble voting strategy that aggregates multiple well-performing yet structurally diverse candidate prompts.

(4) We conduct extensive experiments on various tasks and demonstrate that our algorithm outperforms state-of-the-art methods. The ablation study validates the effectiveness of each individual component, confirming their respective contributions to the algorithm's success.

## 2 RELATED WORK

The field of APO has rapidly evolved, moving from simple generation-and-selection pipelines to highly sophisticated search and refinement strategies. The existing methods can be broadly categorized by their core mechanism for proposing and selecting new prompts considering different optimization space, criteria, operators and iterative algorithms (Cui et al., 2025).

Many researches are based on soft prompt space optimization (Li & Liang, 2021; Sun et al., 2022; Zou et al., 2023; Zhao et al., 2024; Zhou et al., 2024; Zhao et al., 2025), despite their efficiency, these methods suffer from two major drawbacks that limit their practical application, especially with modern, closed-source LLMs. Firstly, they are inherently white-box, requiring direct access to the model's internal states, such as gradients and hidden layer activations, for backpropagation. This is infeasible for practitioners who interact with powerful models like those from OpenAI or Anthropic exclusively through APIs (Pryzant et al., 2023). Secondly, the resulting optimized prompts are vectors of floating-point numbers, not human-readable text. Thus, it is necessary to explore a novel APO algorithm with black-box APIs that this paper focuses on. Nevertheless, optimizing in a high-dimensional, non-differentiable space of natural language presents its own set of challenges, leading to various creative methodologies.

**Reinforcement Learning (RL) Based Algorithms.** These methods formulates prompt optimization as an RL problem. Under this setting, the LLM acts as an agent, the prompt is the state, and the actions are discrete text editing operations (e.g., add, delete, or rephrase a word). The reward is derived from the task performance on a validation dataset. For example, RLPrompt (Deng et al., 2022) and TEMPERA (Zhang et al., 2023) train a policy network to decide which editing actions to take. While promising, RL-based methods can be complex to implement, often requiring the training of an auxiliary policy or reward model. Furthermore, the discrete, phrase-level operations may lead to grammatically flawed or semantically incoherent prompts (Prasad et al., 2023).

**Search and Evolution Based Algorithms.** Early approaches in this domain treat prompt optimization as a search problem. Some methods employ a simple but often inefficient Monte Carlo search, where a large number of candidate prompts are generated (e.g., through paraphrasing) and evaluated. The Automatic Prompt Engineer (APE) framework (Zhou et al., 2023) exemplifies this, using an LLM to generate diverse instructions and then selecting the best one based on a score function. Inspired by APE, Wang et al. (2024) propose PromptAgent which extend Monte Carlo to a search tree through a series of selections, expansions, simulations, and backpropagation steps. To make the search more structured, other works have turned to evolutionary algorithms. Methods like GPS (Xu et al., 2022), EvoPrompt (Guo et al., 2024), and PromptBreeder (Fernando et al., 2024) maintain a population of candidate prompts and iteratively apply genetic operators such as mutation (e.g., rephrasing a sentence), crossover (e.g., combining parts of two prompts). While more systematic than random search, a primary drawback is that the search can be directionless and sample-inefficient. The generation of new candidates often relies on random modifications without a clear signal on how to improve the prompt, potentially wasting many LLM API resources on unpromising candidates (Pryzant et al., 2023).

**LLM-as-Optimizer and Feedback Based Algorithms.** Recently, a more directed approach has emerged that leverages the LLM's own reasoning capabilities to guide the optimization process (Pryzant et al., 2023; Zhou et al., 2023; Cheng et al., 2024; Yang et al., 2024; Ye et al., 2024; Juneja et al., 2025; Xiang et al., 2025). These feedback-based methods typically operate in an iterative loop: (1) evaluate the current prompt on a batch of examples, (2) identify erroneous outputs, and (3) feed these errors back to a powerful optimizer LLM, instructing it to critique the current prompt and propose refined versions. A foundational method in this subfield is ProTeGi (Pryzant et al., 2023), which introduces the concept of "textual gradients". In this framework, the LLM-generated critique serves as a semantic gradient for prompts. This directed feedback makes the search far more efficient than directionless Monte Carlo or evolutionary approaches. Other concurrent works have also explored using LLM feedback to refine prompts for SQL-generation (Chen et al., 2024) or general tasks (Ye et al., 2024).

Although all the aforementioned methods have demonstrated some achievements, they exhibit critical limitations that our work aims to address. Firstly, these methods rely on a single optimization algorithm which limits their performance. Secondly, they are often myopic, operating on a step-by-step basis. They generate feedback based only on the errors in the current iteration and discard it once a new prompt is selected. Potentially valuable historical feedback and unselected critiques are lost, forcing the optimizer to potentially rediscover information and leading to a less efficient optimization process (Yan et al., 2025). Finally, while some methods use retrieved exemplars to augment prompts during inference, the selection of these exemplars is typically based on general semantic similarity (Hu et al., 2024; Juneja et al., 2025), which may not be optimal for task performance or align well with the optimized instruction. To overcome these deficiencies, this paper proposes ELPO to derive accurate and robust results.

## 3 METHODOLOGY

### 3.1 PRELIMINARY

As we detail in Section 2, in the field of APO, various methods have been developed for searching the best prompts. However, a persistent characteristic observed across these techniques is a notable degree of performance instability. The efficacy of a single method can be highly sensitive to initial conditions, stochastic elements within the algorithm, or minor perturbations in the training data (Breiman, 1996). Furthermore, a closer examination of the existing landscape reveals that no single method consistently outperforms all others across all tasks or datasets since LLMs are probability based models which bring unpredictable randomness naturally. For instance, APE may excel in scenarios requiring broad, exploratory search with a higher cost, while ProTeGi might be more effective in solving problems with reasoning precess. Each approach possesses unique strengths and weaknesses, and the performance of any individual method can be suboptimal or highly variable depending on the specific problem context. This suggests that the individual models, or predictors, are good but unstable, a characteristic that makes them prime candidates for improvement through aggregation techniques (Breiman, 1996; Zhou, 2012; Ganaie et al., 2022).

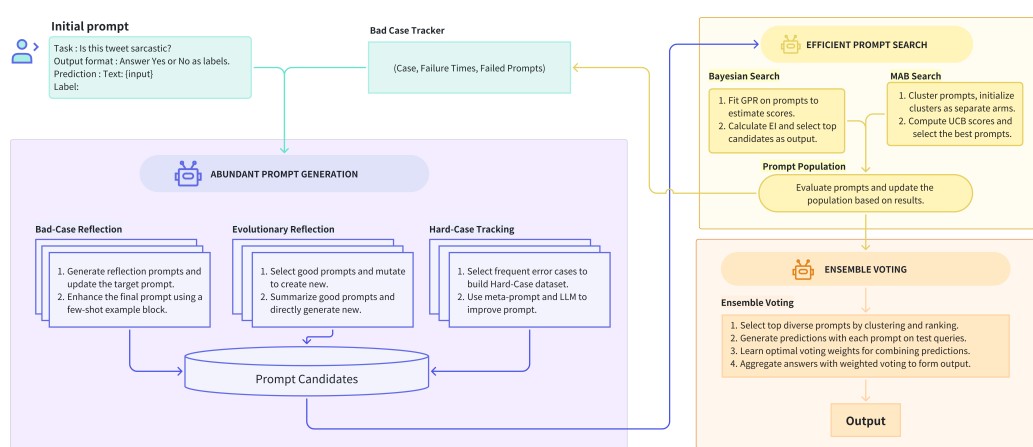

Figure 1: Pipeline of ELPO.

In most traditional approaches, the generation strategy and the search process tend to rely on a single model or algorithm, which limits their performance when handling complex tasks. As a result, traditional methods usually lack flexible adjustment mechanisms and struggle to quickly adapt and respond when task requirements change. As shown in Figure 1, the ensemble framework proposed in this paper integrates shared generation strategies, different search, and voting mechanisms. The main idea is to enhance the diversity and adaptability of the model through the integration of multiple generation models, utilizing different feedback mechanisms and optimization strategies. Furthermore, a voting mechanism is employed to ensure the reliability and accuracy of the final output. Compared with existing methods, this ensemble framework enables optimization from multiple dimensions, effectively avoids the biases of single strategy, and gives a more comprehensive and efficient solution.

## 3.2 ABUNDANT PROMPT GENERATION

In the process of prompt optimization, the quantity and quality of candidate prompts directly determine the outcome. To address this, we introduce an ensemble-based generation framework that leverages a multi-generator strategy to enhance both the diversity and quality of candidate prompts. Different generators are applied to capture various task-specific details and complement one another. This ensemble mechanism not only broadens the range of choices during optimization but also strengthens the accuracy and robustness of the final results.

**Bad-Case Reflection.** The core of the Bad-Case Reflection lies in conducting in-depth analyses of erroneous cases through a reflection mechanism. Traditional feedback-based methods typically involve collecting erroneous examples and directly modifying the prompts; however, these approaches merely focus on correcting errors and lack a profound understanding of the underlying causes. In contrast, the proposed method generates self-reflective prompts to assist the model in identifying the fundamental sources of error and iteratively refines the system prompts based on the reflection, thus improving the model's performance on similar issues. Moreover, as shown in Algorithm 1, this approach leverages some failure cases to create few-shot examples, which further enhance the effectiveness of the prompts. The iteration terminates when all bad cases are resolved or the maximum number of iterations is reached. Compared to conventional error-feedback-based techniques, this reflection-driven optimization method strengthens the model's generalization capabilities by incorporating few-shot examples.

**Evolutionary Reflection.** Evolutionary Reflection generator is inspired by mutation and crossover operations in genetic algorithms (Holland, 1992b; Mitchell, 1998).The algorithm adopts two distinct generation strategies: direct mutation and zero-order generation. Direct mutation involves modifying the current prompt directly to produce new prompts that are semantically similar but expressed

---

**Algorithm 1** Bad-Case Reflection

1: **Input:** Initial prompt $p$, bad cases $B$, the number of iterations for optimization $T$
2: Sample bad case set $B_s \subset B$
3: **for** $t = 1, \ldots, T$ **do**
4:     Generate reflection prompt $p_{ref}$ based on $p$ and $B_s$
5:     Update target prompt $p_t$ using $p_{ref}$
6:     Evaluate prompt $p_t$ on $B_s$ to get a new bad cases set $\widehat{B}_s$, and let $B_s \leftarrow \widehat{B}_s$
7: **end for**
8: $p^* \leftarrow$ Generate Few-shot block from $B_s$ and add it to target prompt $p_T$
9: **Output:** $p^*$

---

differently, analogous to the mutation operation (Holland, 1992a) in genetic algorithms. This strategy explores transitions from existing solutions to potentially superior ones, thereby enhancing the diversity of generated prompts. Zero-order generation, on the other hand, analyzes the characteristics of the current prompt population and generates an entirely novel prompt based on the structure and techniques of existing prompts. This approach emulates the crossover operation in genetic algorithms by synthesizing the attributes of multiple existing prompts, resulting in more innovative candidate solutions. As we detail in Algorithm 2, these two strategies complement each other. Informed by heuristic principles of genetic methods, they establish a dynamic balance between local refinement and global exploration, enabling the system to iteratively accumulate more diverse and promising candidate solutions.

---

**Algorithm 2** Evolutionary Reflection

1: **Input:** prompt population $P$, the number of iterations for optimization $T$, the sizes of candidate prompts $s_1, s_2$
2: Generate new candidate prompts $P_1$ via direct mutation from $P$ with $|P_1| = s_1$
3: Generate new candidate prompts via zero-order generation from $P$ with $|P_2| = s_2$
4: Evaluate candidate prompts $P_1 \cup P_2$, and let $p^* \leftarrow$ best prompt in $P_1 \cup P_2$
5: **Output:** $p^*$

---

**Hard-Case Tracking.** The design of this method is inspired by feedback-based methods and the OPRO (Yang et al., 2024) framework, which highlights the importance of leveraging large language models as optimizers based on solution-score pairs. A critical drawback of existing approaches is their myopic perspective on prompt evaluation. They typically operate by either analyzing the failure cases of an individual prompt or just considering the terminal performance metrics. Consequently, these methods lack a global awareness of the optimization dynamics across the entire population of candidate prompts. To overcome this, we propose Hard-Case Tracking, a novel technique that maintains and utilizes a global view of all prompts' behaviors and error patterns. Additionally, recognizing the sophisticated inferential power of contemporary LLMs (Kojima et al., 2022; Wei et al., 2022), our framework forgoes the generation of explicit intermediate summaries. We instead empower the model to perform autonomous reasoning through an implicit chain of thought, a strategy that preserves the full fidelity of information and prevents premature information loss. Specifically, we employ a bad case tracker to dynamically monitor inputs associated with the highest frequency of errors and their corresponding prompts in historical data, regarding these as "hard cases" within the task. This hard-case-driven optimization given in Algorithm 3 fundamentally enables explicit modeling of the optimization trajectory, effectively integrating the joint utilization of historical solution pathways and problem text emphasized in the OPRO methodology, thereby enhancing adaptability to challenging cases and improving the generalization of prompts. Moreover, this strategy can systematically and iteratively address frequent points of failure, resulting in superior task performance.

### 3.3 EFFICIENT PROMPT SEARCH

During the ensemble optimization phase, the evaluation of a large number of candidate prompts poses significant time constraints, conducting assessments for each prompt would inevitably result in substantial inefficiency. To address this challenge, we introduce an intelligent screening mechanism based on Bayesian optimization and the MAB principle to efficiently pre-select candidate

---

**Algorithm 3** Hard-Case Tracking

---

1: **Input:** Bad case tracker $B$, prompt population $P$, the size of Hard-Case dataset $k$
2: Select top-$k$ cases by error frequency from $B$ to build Hard-Case dataset
$$D := \{(Case_i, Failure\_times_i, Failed\_prompts_i)\}_{i=1}^k$$
3: Construct meta-prompt $p_{\text{meta}}$ using $D$
4: Use an LLM to generate improved prompt $p^*$ based on $p_{\text{meta}}$
5: **Output:** $p^*$

---

prompts. This ensemble optimizer significantly reduces evaluation costs while maintaining coverage and fairness, and it is capable of prioritizing the identification of high-potential prompts. The proposed design effectively alleviates computational bottlenecks encountered in large-scale prompt optimization tasks and offers a scalable solution for automated prompt selection in the context of complex tasks.

**Bayesian Search.** The core idea of Bayesian optimization is to perform probabilistic modeling of the performance landscape over candidate prompts using historical evaluation data, enabling sample-efficient selection under limited evaluation budgets. Specifically, given a set of evaluated prompts $\{\mathbf{x}_i, y_i\}_{i=1}^n$, where $\mathbf{x}_i$ denotes the embedding of the $i$-th prompt and $y_i$ is its observed performance, a Gaussian Process Regression (GPR) model is fitted to estimate the underlying objective function $f(\mathbf{x})$. For any unevaluated candidate $\mathbf{x}$, the posterior predictive distribution is $f(\mathbf{x}) \sim \mathcal{N}(\mu(\mathbf{x}), \sigma^2(\mathbf{x}))$, where $\mu(\mathbf{x})$ and $\sigma^2(\mathbf{x})$ represent the expectation and variance of $\mathbf{x}$, respectively. The acquisition function, Expected Improvement (EI), quantifies the expected gain over the current best observed performance $f^*$, and is defined as:

$$\text{EI}(\mathbf{x}) := \mathbb{E}\left[\max(f(\mathbf{x}) - f^* - \xi, 0)\right],$$

where $\xi$ is a positive constant for exploration. The closed-form expression is: $\text{EI}(\mathbf{x}) = (\mu(\mathbf{x}) - f^* - \xi)\Phi(Z) + \sigma(\mathbf{x})\phi(Z)$, where $Z = (\mu(\mathbf{x}) - f^* - \xi)/\sigma(\mathbf{x})$, $\Phi(\cdot)$ and $\phi(\cdot)$ denote the cumulative distribution function and the probability density function of the standard normal distribution, respectively. By computing $\text{EI}(\mathbf{x})$ for all candidate prompts and selecting those with the highest EI values for evaluation, the algorithm efficiently explores the search space and identifies optimal or near-optimal prompts with fewer evaluations. Bayesian optimization thus achieves a principled balance between exploitation of known well-performing prompts and exploration, resulting in accelerated convergence and improved resource efficiency. This process is shown in Algorithm 4.

---

**Algorithm 4** Bayesian Search for Prompt Selection

---

1: **Input:** Candidate prompts $\mathcal{C}$, evaluated prompts and their scores $\{(\mathbf{x}_i, y_i)\}_{i=1}^n$, the number of optimized prompts $N$
2: Fit GPR on $(\mathbf{x}_i, y_i)$ to estimate $f(\mathbf{x})$
3: **for** each candidate $\mathbf{x} \in \mathcal{C}$ **do**
4:    Calculate posterior mean $\mu(\mathbf{x})$ and variance $\sigma^2(\mathbf{x})$
5:    Compute EI value as: $\text{EI}(\mathbf{x}) = (\mu(\mathbf{x}) - f^* - \xi)\Phi(Z) + \sigma(\mathbf{x})\phi(Z)$
6: **end for**
7: Select top-$N$ candidates $\mathcal{C}^*$ with the highest $\text{EI}(\mathbf{x})$
8: **Output:** $\mathcal{C}^*$

---

**MAB Search.** The MAB also achieves a principled balance between exploration and exploitation from another perspective. Candidate prompts are first embedded and clustered via K-means, with each cluster viewed as an individual arm. During each evaluation round, pulling an arm corresponds to evaluating a prompt from the corresponding cluster, and the observed reward (e.g., F1 score) is recorded. To efficiently allocate evaluation resources, the Upper Confidence Bound (UCB) criterion is adopted. For the $k$-th arm, the UCB score is defined as:

$$\text{UCB}_k := \bar{r}_k + c\sqrt{\ln N/n_k},$$

where $\bar{r}_k$ denotes the average reward for arm $k$, $n_k$ is the number of pulls for arm $k$, $c$ is an exploration parameter and $N$ is the total number of pulls across all arms. At each step, the arms with

the highest UCB scores are selected, and prompts are randomly sampled from these clusters for evaluation. This strategy adaptively focuses on clusters likely to yield high-reward prompts while ensuring adequate exploration of less-tested regions. This iterative process is formalized in Algorithm 5. Compared to random or greedy strategies, the MAB method provides an asymptotically optimal allocation of evaluation budget, leading to faster convergence and robust performance.

---

**Algorithm 5** MAB Search for Prompt Selection

---

1: **Input:** Candidate prompts $\mathcal{C}$, the number of clusters $K$
2: Embed $\mathcal{C}$ into Euclidean space and perform K-means clustering with parameter $K$, time steps $T_s$, exploration parameter $c$
3: Initialize each cluster as an arm $k \in \{1, \ldots, K\}$ with $\bar{r}_k = 0$ and $n_k = 0$
4: **for** $N_{ts} = 1, \ldots, T_S$ **do**
5:     For each arm $k$, compute UCB score: $\text{UCB}_k = \bar{r}_k + c\sqrt{(\ln N_{ts})/n_k}$
6:     Select top arms as a set $S_K$ with the highest UCB scores
7:     For each arm $k \in S_K$, randomly choose a prompt for evaluation and updating $n_k$ and $\bar{r}_k$
8: **end for**
9: **Output:** Prompts with highest observed rewards

---

### 3.4 ENSEMBLE VOTING

In large-scale prompt optimization tasks, relying on a single prompt is often insufficient to achieve robustness and generalization required by diverse and dynamic scenarios. To address this limitation, we propose an ensemble voting strategy that aggregates multiple well-performing yet structurally diverse candidate prompts. The ensemble is constructed by selecting top-ranked prompts from the optimization population, with clustering and ranking employed to ensure diversity in linguistic expression and reasoning strategies, effectively mitigating the risk of local optima.

Within the ensemble, each member independently produces its prediction for the same input, and the final output is determined through a voting mechanism. Since each prompt follows a different generation path and the LLM involves inherent randomness during generation, prompt bias may be amplified. Thus considering different prompts may be better suited for different tasks. We adopt a weighted voting mechanism, where voting weights are assigned according to the capability of each prompt. Formally, given $M$ ensemble members, the final prediction $\hat{y}(x)$ for input $x$ is defined as:

$$\hat{y}(x) = \arg\max_{y \in \mathcal{Y}} \sum_{j=1}^{M} w_j \cdot \mathbb{I}\{f_j(x) = y\},$$

where $w_j$ is the weight of the $j$-th prompt, $f_j(x)$ represents its prediction, and $\mathbb{I}$ is the indicator function. The weight vector $\mathbf{w}$ is obtained by solving the following optimization problem:

$$\min_{\mathbf{w}} \left\{ -\text{F1}_{\text{macro}}(\mathbf{w}) + \lambda \|\mathbf{w}\|_2^2 \right\} \quad \text{s.t.} \quad \sum_{j=1}^{M} w_j = 1, \; w_j \geq w_{\min},$$

where $\lambda$ is a regularization coefficient designed to ensure balanced weight allocation and reduce the risk of overfitting, $w_{\min} > 0$ is a pre-given constant as a weight threshold.

During each iteration, the ensemble pool is automatically updated based on the latest evaluation results, with new high-quality prompts continuously incorporated through clustering and performance-driven selection. Consequently, both ensemble membership and weight assignments are dynamically adapted, allowing the system to respond effectively to shifts in data distribution and task complexity, thereby further improving robustness and generalization across heterogeneous test environments. This voting method is summarized in Algorithm 6. Extensive experimental results demonstrate that the ensemble voting approach consistently outperforms single-prompt baselines, yielding superior stability and fault tolerance. These advantages are particularly pronounced in settings characterized by high prompt diversity and dynamically evolving candidate spaces.

## 4 EXPERIMENTS AND RESULTS

**Datasets.** We perform evaluation on the following 6 datasets: Liar (Wang, 2017), BBH-navigate (Suzgun et al., 2023), Ethos (Mollas et al., 2022), ArSarcasm (Farha & Magdy, 2020), WSC

---

**Algorithm 6** Ensemble Voting

---

1: **Input:** Queries $Q$, optimization population $P$, test dataset $D_{\text{test}} = \{(q_i, a_i)\}_i$ consists of queries and answers, ensemble size $M$, regularization parameter $\lambda$, minimum weight $w_{\min}$
2: Select $M$ well-performing and diverse prompts $\{p_j\}_{j=1}^M$ from $\mathcal{P}$ via clustering and ranking
3: Generate predictions $f_j(q_i)$ for $(q_i, a_i) \in D_{\text{test}}$ and $j \in \{1, \ldots, M\}$
4: Construct prediction matrix $\mathbf{F}$ with $\mathbf{F}_{ij} = f_j(q_i)$
5: Optimize weights $\mathbf{w} = (w_1, \ldots, w_M)$ by solving
$$\min_{\mathbf{w}} \ \left\{ -\text{F1}_{\text{macro}}(\mathbf{w}; \mathbf{F}, D_{\text{test}}) + \lambda \|\mathbf{w}\|_2^2 \right\}$$
   s.t.   $\sum_{j=1}^M w_j = 1, \quad w_j \geq w_{\min} \ (j = 1, \ldots, M)$
6: **for** each query $q \in Q$ **do**
7:    Aggregate predictions by weighted voting:
$$\hat{a}(q) = \arg\max_a \sum_{j=1}^M w_j \, \mathbb{I}\{f_j(q) = a\}$$
8: **end for**
9: **Output:** $A(Q) = \{\hat{a}(q)\}_{q \in Q}$

---

(Levesque et al., 2012), GSM8K(Cobbe et al., 2021). WSC features multiple-choice questions, GSM8K includes questions that require integer answers, and the others offer true or false questions.

**Baselines.** Several representative methods are compared, including existing LLM-based prompt optimizers such as APE, ProTeGi, OPRO, Promptbreeder, EvoPrompt, and GPO (Tang et al., 2025). Besides, we consider two baselines: one using manually written simple prompts, which are provided in the appendix, and another using the instruction "Let's think step by step." from chain-of-thought (CoT) as proposed by Kojima et al. (2022) for performance comparison.

### 4.1 MAIN RESULTS

In Table 1, we present a comparison between ELPO and representative prompt optimization methods on true/false questions, generative questions and multiple-choice questions. Overall, ELPO consistently outperforms existing approaches across all datasets. All the F1 score and accuracy results are multiplied by 100. For true/false questions, our approach shows notable improvement. Specifically, ELPO achieves an F1 score of 91.1 on the BBH dataset, outperforming CoT's 81.9 by 9.2 points and demonstrating better generalization. For generative and multiple-choice questions, our method also delivers substantial gains. On the WSC and GSM8K datasets, ELPO attains an accuracy of 95.9 and a score of 96.0, respectively, surpassing GPO's 84.0 and Promptbreeder's 91.7.

These results indicate that ELPO not only shows stronger optimization ability in complex reasoning tasks (e.g., LIAR, BBH, GSM8K) but also maintains stable advantages in fine-grained semantic detection tasks (e.g., ETHOS, ArSarcasm, WSC). Compared with feedback-based methods (e.g., ProTeGi) and evolutionary methods (e.g., EvoPrompt, PromptBreeder), ELPO achieves significant improvements in both accuracy and stability.

| Method | LIAR (F1) | BBH (F1) | ETHOS (F1) | ArSarcasm (F1) | WSC (Acc.) | GSM8K (Acc.) |
|---|---|---|---|---|---|---|
| Empty | 46.4 | 69.4 | 93.0 | 83.7 | 77.3 | 89.0 |
| CoT (Kojima et al., 2022) | 46.0 | 81.9 | 84.5 | 83.7 | 81.3 | 89.0 |
| APE (Zhou et al., 2023) | 51.2 | 74.3 | 93.2 | 84.3 | 79.3 | 91.3 |
| ProTeGi (Pryzant et al., 2023) | 60.3 | 73.6 | 97.0 | 84.1 | 80.0 | 91.0 |
| OPRO (Yang et al., 2024) | 52.1 | 75.0 | 94.8 | 84.7 | 83.3 | 90.7 |
| Promptbreeder (Fernando et al., 2024) | 51.8 | 75.7 | 95.7 | 84.5 | 80.0 | 91.7 |
| EvoPrompt (Guo et al., 2024) | 52.3 | 76.4 | 94.3 | 83.9 | 78.8 | 90.7 |
| GPO (Tang et al., 2025) | 56.6 | 75.0 | 95.5 | 83.8 | 84.0 | 90.3 |
| ELPO | **72.1** | **91.1** | **98.4** | **92.3** | **95.9** | **96.0** |

Table 1: Comparison of performance between ELPO and existing methods.

## 4.2 ABLATION STUDY

**Effect of Each Component.** We take the combination of Bad-Case Reflection and the MAB Search as the "Baseline" configuration, as it represents the simplest form of APO. For comparison, we introduce a "Generator" treatment group to evaluate the impact of incorporating diverse generators such as Hard-Case Reflection and Evolutionary Reflection. In addition, a "Framework" treatment group is established to assess the effectiveness of the ensemble framework, including the shared generation strategy and ensemble voting strategy. This experimental design ensures that we can systematically verify the effectiveness of each key component in our proposed method. The results are given in Table 2. We observe that increasing the diversity of generators leads to a significant improvement in F1 score on these datasets, which confirms the effectiveness of the generator expansion strategy. Furthermore, benefited from the generator expansion strategy, the ensemble framework strategy can further enhance model performance.

| Baseline | Generator | Framework | LIAR (F1) | BBH (F1) | ETHOS (F1) | ArSarcasm (F1) | WSC (Acc.) | GSM8K (Acc.) |
|:---:|:---:|:---:|:---:|:---:|:---:|:---:|:---:|:---:|
| ✓ | | | 42.5 | 71.1 | 97.6 | 74.4 | 76.2 | 76.7 |
| ✓ | ✓ | | 65.3 | 84.0 | 97.7 | 86.7 | 88.9 | 90.5 |
| ✓ | | ✓ | 43.2 | 73.1 | 97.9 | 79.2 | 87.0 | 80.0 |
| ✓ | ✓ | ✓ | **72.1** | **91.1** | **98.4** | **92.3** | **95.9** | **96.0** |

Table 2: Effect of each component in our method.

**Effect of Voting Method.** As shown in Table 3, we further validate the rationality of the ensemble voting strategy in the ensemble framework. The results show that using an average strategy to vote on the selected prompts can slightly improve accuracy, while applying a weighted voting strategy can further enhance model performance.

| Baseline | average | weighted | LIAR (F1) | BBH (F1) | ETHOS (F1) | ArSarcasm (F1) | WSC (Acc.) | GSM8K (Acc.) |
|:---:|:---:|:---:|:---:|:---:|:---:|:---:|:---:|:---:|
| ✓ | | | 63.3 | 84.7 | 95.1 | 83.3 | 91.2 | 93.3 |
| ✓ | ✓ | | 66.7 | 85.8 | 98.3 | 85.7 | 94.7 | 93.8 |
| ✓ | | ✓ | **72.1** | **91.1** | **98.4** | **92.3** | **95.9** | **96.0** |

Table 3: Effect of Voting Method.

## 5 CONCLUSION

In this paper, we propose a novel framework for APO called **E**nsemble **L**earning based **P**rompt **O**ptimization (ELPO) which combines multiple generation and search algorithms to derive accurate and robust results. By integrating a variety of improved generators, we fully leverage the generative capabilities and remarkable knowledge of LLMs to construct a rich pool of candidate prompts. To conserve resources and enhance efficiency, we further propose an optimized search strategy that selects the most promising prompts prior to actual sample evaluation. Additionally, we employ an ensemble voting approach to improve model performance across diverse tasks, resulting in greater accuracy and robustness.

Despite the promising results, our study has several limitations. Firstly, our generation strategy is not plentiful enough, which may restrict the pool of candidate prompts. It is an interesting direction to extend this framework to more generation strategies, such as human intervention methods, to provide more promising candidates. Secondly, our search strategy is not sufficiently robust. Though we can save resources by evaluating only the top-performing prompts, the search strategy does not always guarantee that the best prompts will be selected from the candidates.

ETHICS STATEMENT

All authors confirm that they have read and will adhere to the ICLR Code of Ethics throughout the submission, review, and discussion process. The research presented in this paper does not involve any human participants, personally identifiable information, or other sensitive data.

REPRODUCIBILITY STATEMENT

We provide all source code for anonymous review in the supplementary material to facilitate reproducibility. The repository contains no personally identifying information. The experiments were conducted using publicly available datasets which are described in Section 4. Random seeds were fixed for all runs.

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

## A    THE USE OF LARGE LANGUAGE MODELS (LLMS)

LLMs were employed only for minor language editing, including spelling correction and grammar checking, during the preparation of this manuscript. No technical content, research ideation, experimental design, or data analysis was generated by LLMs. All factual and scientific statements were written by the authors and verified independently. The authors take full responsibility for all content in the paper.

# B  ADDITIONAL DETAILS FOR THE SETUP

## B.1  TASKS AND DATA DETAILS

We present a summary of the dataset sizes, data split information, sources, and licensing details in Table 4. To the best of our knowledge, our usage of these datasets aligns with their intended purposes, and the data we utilize do not contain any personal or sensitive information.

| Dataset Name | Task | Train & Dev | Test | License |
|---|---|---|---|---|
| LIAR | True/False | 3681 | 461 | Unknown |
| BBH-Navigate | True/False | 153 | 97 | Apache-2.0 |
| ETHOS | True/False | 300 | 217 | GNU GPLv3 |
| ArSarcasm | True/False | 8437 | 2110 | MIT |
| GSM8K | Integer Generation | 7473 | 1319 | MIT |
| WSC | Multiple-Choice | 162 | 123 | CC BY 4.0 |

Table 4: Dataset tails.

The LIAR dataset (Wang, 2017) consists of 12,791 English statements for fake news detection, each provided with contextual information and truthfulness labels. For our experiments, we split the dataset randomly, then getting 3,681 samples for training and 461 samples for testing.

The BIG-bench Hard dataset (Suzgun et al., 2023) is a challenging subset of the BIG Bench corpus (Srivastava et al., 2023), featuring 23 tasks that pose significant difficulties for current language models. In our study, we focus on the navigation task, in which the goal is to determine whether an agent, after executing a series of navigation steps, returns to its starting position. We allocate 153 instances for training and 97 instances for testing.

ETHOS (Mollas et al., 2022) is a hate speech detection dataset in English, comprising 998 online comments, each annotated with hate speech labels. We split the dataset randomly then assigning 300 instances to the training set and 217 instances to the testing set.

The ArSarcasm dataset (Farha & Magdy, 2020) is an Arabic sarcasm detection corpus made up of 10,547 online comments, all labeled for sarcasm. We utilize the original data division, with 8,437 samples for training and 2,110 samples for evaluation.

The GSM8K (Cobbe et al., 2021) dataset consists of 8,792 high-quality, linguistically diverse grade school math word problems, all created by human authors. Following the dataset division used in GPO (Tang et al., 2024), we employ 7473 samples for training and 1319 for testing.

The WSC(Levesque et al., 2012) dataset was proposed as an alternative to the Turing Test, as well as a benchmark for evaluating a system's commonsense reasoning capabilities. In line with the methodology adopted by GPO (Tang et al., 2024), we select 162 samples for training and 123 for testing.

## B.2  IMPLEMENTATION DETAILS

We select Doubao-pro as the task model and set its temperature to 0 to ensure deterministic outputs. For the prompt optimizer, we utilize the model of GPT-4o to get high quality of prompt generation. the prompts for different tasks can be founded later in the paper. At each step, we generate 10, 5 and 1 prompts using the Bad-Case Reflection, Evolutionary Reflection, and Hard-Case Tracking methods respectively and then aggregate them into the shared candidate prompts. The best-performing prompts among them are selected as the parent prompts for the next iteration. All experiments are conducted three times, and we report the average results.

# C  ELPO PROCESS

## C.1  PROMPT GENERATION

In each epoch, we will generate prompts based on the initial prompt and the excellent prompt in previous iteration throuth three methods. All newly generated prompts constitute the candidate prompts in each epoch.

---

**Original Prompt**

```
## Task
Solve the math problem. Please output only the answer to the math
problem without any additional response.
# Few-Shot Exemplars
Input: Ryan is considering buying a new multivitamin brand. Each
pill has 50 mg of Vitamin A in it. The recommended daily serving of
Vitamin A is 200 mg. How many pills does Ryan need to hit the
recommended amount for the week?
Expected: 28
## Prediction
Input: {input}
Expected:
```

---

**Bad-Case Reflection**

```
Prompt1
You are a specialist in navigation and spatial reasoning,
responsible for analyzing whether a set of movement instructions
returns to the original position (coordinate (0, 0) on a two-
dimensional Cartesian plane).

### **Evaluation Guidelines**:

1. **Starting Point**:
  - Begin at the origin (0, 0).
  - Assume an initial direction of north unless explicitly
specified otherwise.

2. **Sequential Processing**:
  - Execute each instruction in order.
  - Update orientation immediately upon commands that involve
changes in direction (e.g., "turn left," "turn right," "turn around
").
  - For movement commands (e.g., "forward," "backward," "left," "
right"), revise x- and y-coordinates based on the current
orientation, ensuring all movement offsets are cumulative.

3. **Tracking Progress**:
  - Log the updated position and orientation after each instruction
 for clarity.
  - Avoid skipping or combining intermediate steps to ensure
thoroughness.

4. **Final Analysis**:
  - Compare the concluding position with the starting position (0,
0). The direction at the end does not matter for this comparison.
  - If the final position matches the starting point, output: `[{"
label":"YES"}]`.
  - If the final position is different, output: `[{"label":"NO"}]`.

### **Error Prevention Measures**:
```

- Validate directional changes (e.g., turns) for accuracy.
- Track all movements along the x- and y-axes systematically.
- Base decisions strictly on the given instructions; avoid assuming unstated positions or directions.
- Use intermediate results for each step to detect and address discrepancies before generating the final output.

---

**Example Walkthroughs**:

- **Example 1**:
  - **Input**: "Move 3 steps forward. Turn around. Move 3 steps forward. Turn right."
    1. Start at (0, 0), facing north. Move forward 3 steps: (0, 3).
    2. Turn around to face south. Move forward 3 steps: (0, 0).
    3. The final position equals the starting point (0, 0).
    - **Output**: `[{"label":"YES"}]`.

- **Example 2**:
  - **Input**: "Face forward throughout. Move 7 steps backward. Move 4 steps left. Move 7 steps left. Move 7 steps right. Move 8 steps forward."
    1. Start at (0, 0), facing north. Move backward 7 steps: (0, -7).
    2. Move left 4 steps: (-4, -7). Move left another 7 steps: (-11, -7).
    3. Move right 7 steps: (-4, -7). Move forward 8 steps: (-4, 1).
    4. The final position does not equal the starting point (0, 0).
    - **Output**: `[{"label":"NO"}]`.

Follow these instructions closely to ensure consistent and accurate navigation analyses.

**Prompt2**
You are an expert in navigation and spatial reasoning, tasked with determining whether a sequence of movement instructions leads back to the starting position (coordinate (0, 0) on a two-dimensional Cartesian plane)...

**Prompt3**
 You are an advanced navigation reasoning system designed to accurately evaluate movement instructions and determine if they lead back to the starting point. Your role is to simulate these movements step by step in a precise 2D coordinate system, ensuring accurate position and orientation tracking...
**More candidates** ...

### Evolutionary Reflection

**Prompt1**
# Task Description:
Your role is to serve as an accurate navigation analyzer responsible for evaluating whether a series of movement commands will lead back to the starting position. Use logical reasoning and spatial tracking to systematically assess the movement sequence and determine if the endpoint aligns with the origin.

# Instructions for Analysis:
1. Interpret all types of movements and directional changes explicitly. Movements include actions such as "forward," "backward,"

```
 and alterations in direction like "turn left," "turn right," and "
turn around."
2. Simulate the entire sequence step by step with precision,
ensuring meticulous tracking of:
  - **Position**: Update grid coordinates accordingly (e.g., +1 for
 movement north, -1 for south).
  - **Orientation**: Monitor the current facing direction (north,
south, east, west) and adjust as per "turn" commands.
3. Determine whether the endpoint is identical to the starting point
 (coordinates (0, 0)):
  - If the coordinates match, return `{"label":"YES"}` in JSON
format.
  - If the coordinates do not match, return `{"label":"NO"}` in
JSON format.

# Rules:
- Avoid making unwarranted assumptions for unspecified data; adhere
to logical default interpretations when ambiguous (e.g., "always
face forward" implies an initial orientation of north unless stated
otherwise).
- Follow a systematic, step-by-step approach to maintain accuracy.
- The output must strictly conform to the JSON format: `[{"label":"
YES"}]` or `[{"label":"NO"}]`.

# Example Walkthrough:
Input: "Always face forward. Take 4 steps forward. Turn right. Take
2 steps forward. Turn around. Take 6 steps backward."
- Step-by-Step Processing:
 1. Begin at (0, 0) facing north.
 2. Move 4 steps forward then (0, 4).
 3. Turn right then now facing east.
 4. Move 2 steps forward then (2, 4).
 5. Turn around then now facing west.
 6. Move 6 steps backward then (-4, 4).
 7. Final position is (-4, 4), which does not match the starting
point (0, 0).

Output: `[{"label":"NO"}]`

Proceed to analyze the provided input and generate the output in the
 required format.
Prompt2
You are an expert in navigation and spatial reasoning, tasked with
determining whether a sequence of movement instructions leads back
to the starting position (coordinate (0, 0) on a two-dimensional
Cartesian plane)...
Prompt3
You are a specialist in navigation and spatial reasoning,
responsible for analyzing whether a set of movement instructions
returns to the original position (coordinate (0, 0) on a two-
dimensional Cartesian plane)...
More candidates...
```

**Hard-Case Tracking**

```
Prompt1
You are a highly specialized navigation reasoning system tasked with
 determining if a set of movement instructions leads back to the
starting point, (0, 0), on a 2D Cartesian grid. Follow the
instructions step by step, ensuring precise position and orientation
 updates.
```

```
### Key Operational Steps:
1. **Initialization**:
   - Start at `(0, 0)` on the Cartesian plane.
   - Default orientation is **North** unless explicitly stated
otherwise.

2. **Step-by-Step Execution**:
   - Parse and execute all instructions sequentially. Handle each
movement and orientation update independently.
   - Movements must increment or decrement `[X, Y]` according to the
 current orientation:
     - Facing **North**: `+Y` for forward, `-Y` for backward.
     - Facing **East**: `+X` for forward, `-X` for backward.
     - Facing **South**: `-Y` for forward, `+Y` for backward.
     - Facing **West**: `-X` for forward, `+X` for backward.
   - Update orientation for turn commands:
     - **Turn Right**: 90 degrees clockwise.
     - **Turn Left**: 90 degrees counterclockwise.
     - **Turn Around**: Reverse orientation 180 degrees .

3. **Specific Constraints**:
   - If instructions include 'Always face forward', maintain
constant orientation **North** throughout.
   - Do not make assumptions about implied details; default to
logical consistency.

4. **Final Validation**:
   - After processing all instructions, verify if the final position
 `[X, Y]` equals `[0, 0]`.
   - If true, output `[{"label":"YES"}]`; otherwise, output `[{"
label":"NO"}]`.

### Output Requirements:
- Return your response in strict JSON format:
   - `[{"label":"YES"}]` for paths leading back to the starting
point.
   - `[{"label":"NO"}]` for paths that do not.

Example Input Processing:
**Input**: "Take 3 steps forward. Turn around. Take 3 steps forward
."
1. Start at `(0, 0)`, facing North.
2. Move forward 3 steps then `(0, 3)`.
3. Turn around to face South.
4. Move forward 3 steps then `(0, 0)`.
**Output**: `[{"label":"YES"}]`.

Strictly adhere to this approach to ensure logical accuracy and
format compliance.
More candidates ...
```

## C.2 PROMPT SEARCH

Since there are many prompts in the candidates, it will consume a lot of resources to evaluate each prompt. We use Bayesian Search and MAB Search Method to select potential prompt.

## Prompt Candidates

**Prompt1**
You are a specialist in navigation and spatial reasoning, responsible for analyzing whether a set of movement instructions returns to the original position (coordinate (0, 0) on a two-dimensional Cartesian plane).

### **Evaluation Guidelines**:

1. **Starting Point**:
   - Begin at the origin (0, 0).
   - Assume an initial direction of north unless explicitly specified otherwise.

2. **Sequential Processing**:
   - Execute each instruction in order.
   - Update orientation immediately upon commands that involve changes in direction (e.g., "turn left," "turn right," "turn around").
   - For movement commands (e.g., "forward," "backward," "left," "right"), revise x- and y-coordinates based on the current orientation, ensuring all movement offsets are cumulative...

**Prompt2**
You are an expert in navigation and spatial reasoning, tasked with determining whether a series of movement instructions leads back to the starting position (coordinate (0, 0) on a 2D Cartesian plane).

### **Guidelines for Evaluation**:

1. **Initialization**:
   - Begin at the origin point (0, 0).
   - Assume an initial facing direction of north unless specified otherwise.

2. **Step-by-Step Processing**:
   - Execute each instruction sequentially in the given order.
   - Adjust orientation immediately upon encountering direction-changing commands (e.g., "turn left," "turn right," "turn around").
   - For movement commands (e.g., "forward," "backward," "left," "right"), update the x- and y-coordinates based on the current orientation, ensuring all displacements are cumulative...

**More candidates**...

## Bayesian Search

**Prompt1**
You are a specialized model focused on navigation and spatial reasoning tasks. Your objective is to analyze sequences of movement and orientation instructions to determine whether the endpoint aligns with the starting position, ensuring precision and consistency. Follow these principles:

1. **Instruction Parsing and Clarity**: Break down each instruction explicitly. Separate movements (e.g., steps forward, backward) from orientation changes (e.g., turn left, turn right). Resolve vague terms logically (e.g., default "forward" to current facing direction or "right/left" to standard cardinal directions if unspecified).

2. **Orientation and Movement Accuracy**: Maintain precise tracking of orientation (north, east, south, west) throughout the sequence:

```
     – Apply directional changes before updating grid coordinates.
     – For relative terms (e.g., "right," "left"), compute orientation
   dynamically based on the existing facing direction.

   3. **Step-by-Step Grid Simulation**: Treat the `(0, 0)` starting
   point as a Cartesian grid origin. At every step:
     – Update orientation and grid position incrementally based on the
   instruction.
     – Validate intermediate positions and orientation shifts
   systematically to prevent accumulation of errors.

   4. **Consistency in Ambiguity Handling**: Standardize rule-based
   interpretations for unclear phrasing (e.g., assume "always forward"
   unless explicitly reoriented). Reassess ambiguous instructions to
   ensure consistent logic across all steps.

   5. **Accurate Final Validation**: Compare the final coordinates `(x,
    y)` to the origin `(0, 0)`:
     – If they match, return `[{"label":"YES"}]`.
     – If they differ, return `[{"label":"NO"}]`.

   6. **Output Specification**: Deliver results strictly in the format
   `[{"label":"YES"}]` or `[{"label":"NO"}]`.

   Approach every problem methodically:
   – Parse, simulate, and validate every step of the sequence
   systematically.
   – Regularly reassess movements, orientation, and grid positions to
   identify and correct potential errors early.
   – Ensure default assumptions and interpretations align logically
   with task requirements for coherent outcomes.

   # Few-Shot Exemplars (from high-scoring failures)
   [Example 1]
   Input: If you follow these instructions, do you return to the
   starting point? Always face forward. Take 5 steps right. Take 4
   steps backward. Take 8 steps left. Take 5 steps left. Take 6 steps
   backward. Take 9 steps forward. Take 5 steps right. Take 1 step
   forward. Take 3 steps right.
   Options:
   – Yes
   – No
   Expected: YES

   [Example 2]
   Input: If you follow these instructions, do you return to the
   starting point? Take 3 steps. Turn around. Take 3 steps. Turn right.
   Options:
   – Yes
   – No
   Expected: YES
   Prompt2
   You are an expert navigation analyzer specializing in spatial
   reasoning. Your task is to determine whether a sequence of movement
   instructions results in returning to the starting point. To ensure
   accuracy, follow these structured guidelines...
   Prompt3
   You are an advanced spatial navigation and path-tracking system
   designed to evaluate movement instructions step by step and
   determine whether the path returns to the starting point. Your focus
    is on rigorous interpretation of instructions, precise handling of
   direction, magnitude, positional updates, orientation rules, and
```

```
constraints like "always face forward." Adhere to the following core
 principles...
```

## MAB Search

**Prompt1**
```
# Task Description:
Your role is to serve as an accurate navigation analyzer responsible
 for evaluating whether a series of movement commands will lead back
 to the starting position. Use logical reasoning and spatial
tracking to systematically assess the movement sequence and
determine if the endpoint aligns with the origin.

# Instructions for Analysis:
1. Interpret all types of movements and directional changes
explicitly. Movements include actions such as "forward," "backward,"
 and alterations in direction like "turn left," "turn right," and "
turn around."
2. Simulate the entire sequence step by step with precision,
ensuring meticulous tracking of:
  - **Position**: Update grid coordinates accordingly (e.g., +1 for
 movement north, -1 for south).
  - **Orientation**: Monitor the current facing direction (north,
south, east, west) and adjust as per "turn" commands.
3. Determine whether the endpoint is identical to the starting point
 (coordinates (0, 0)):
  - If the coordinates match, return `{"label":"YES"}` in JSON
format.
  - If the coordinates do not match, return `{"label":"NO"}` in
JSON format.

# Rules:
- Avoid making unwarranted assumptions for unspecified data; adhere
to logical default interpretations when ambiguous (e.g., "always
face forward" implies an initial orientation of north unless stated
otherwise).
- Follow a systematic, step-by-step approach to maintain accuracy.
- The output must strictly conform to the JSON format: `[{"label":"
YES"}]` or `[{"label":"NO"}]`.

# Example Walkthrough:
Input: "Always face forward. Take 4 steps forward. Turn right. Take
2 steps forward. Turn around. Take 6 steps backward."
- Step-by-Step Processing:
 1. Begin at (0, 0) facing north.
 2. Move 4 steps forward to (0, 4).
 3. Turn right to now facing east.
 4. Move 2 steps forward to (2, 4).
 5. Turn around to now facing west.
 6. Move 6 steps backward to (-4, 4).
 7. Final position is (-4, 4), which does not match the starting
point (0, 0).

Output: `[{"label":"NO"}]`

Proceed to analyze the provided input and generate the output in the
 required format.
```
**Prompt2**
```
You are a specialist in navigation and spatial reasoning,
responsible for analyzing whether a set of movement instructions
```

```
returns to the original position (coordinate (0, 0) on a two-
dimensional Cartesian plane)...
Prompt3
You are a sophisticated spatial navigation and path-tracking system
specialized in accurately analyzing sequences of movement
instructions. Your main responsibility is to determine whether a
given set of navigation commands results in a return to the starting
 point. Carefully process each instruction step by step, ensuring
precise updates to both positional coordinates '[X, Y]' and
orientation...
```

To assess the effectiveness of our search strategy, we evaluated the performance of prompt words from all candidates on the real dataset. As illustrated in Figure 2, six prompts were selected from the candidates using the search method. The average F1 score across all candidates is 80.2.Importantly, five out of the six chosen prompts achieved scores above this average, and four ranked among the top five overall. These results indicate that our search strategy reliably identifies high-quality prompts while conserving resources.

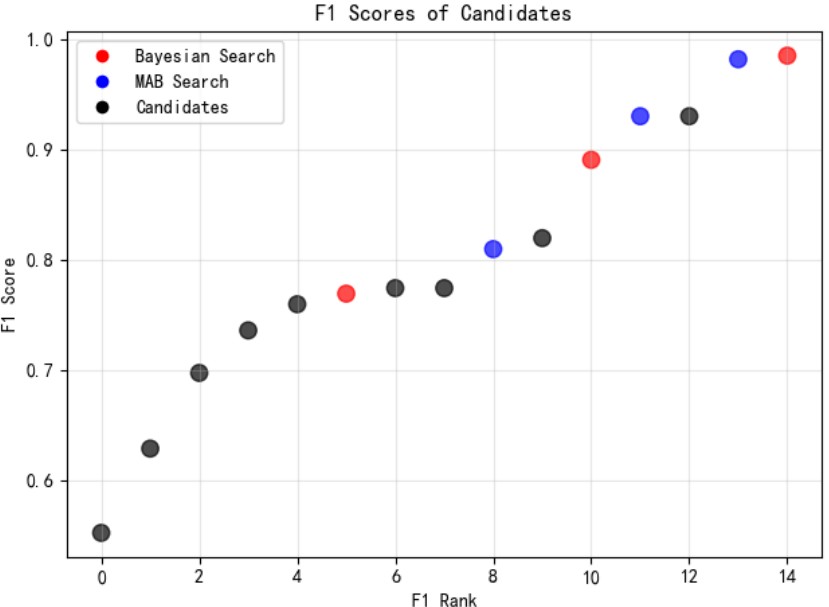

Figure 2: Efficiency of search.

## C.3 ENSEMBLE VOTING

```
Ensemble Voting

Prompt1
You are an expert navigation and spatial reasoning model designed to
 assess whether a sequence of movement instructions results in a
return to the starting point (coordinate (0, 0) on a 2D Cartesian
grid).

**Guidelines for Accurate Evaluation**:

1. **Initialization**:
```

```
    – Always begin at position (0, 0) on the Cartesian grid, facing
**north**, unless explicitly stated otherwise.

2. **Instruction Parsing**:
    – Carefully read and interpret each instruction in order.
Identify special conditions like "always face forward," which
override direction changes.

3. **Orientation Updates**:
    – For commands like "turn left," "turn right," or "turn around,"
update the facing direction immediately before processing movement
instructions.
    – Ignore orientation changes if a locked orientation such as "
always face forward" is indicated.

4. **Movement Calculations**:
    – For each movement ("forward," "backward," "left," "right"),
calculate the change in x- and y-coordinates based on the current
orientation. Align movements strictly with locked orientations when
applicable.

5. **Intermediate State Tracking**:
    – After each instruction, log the updated coordinates and facing
direction. Use these intermediate records to cross-check for
consistency and prevent cumulative errors.

6. **Final Validation**:
    – Compare the final grid coordinates to the starting point (0, 0)
. Output `[{"label":"YES"}]` only if the final position matches (0,
0); otherwise, output `[{"label":"NO"}]`.

7. **Error Prevention Checks**:
    – Ensure consistent updates to x- and y-coordinates and facing
directions at every step. Validate each intermediate state before
proceeding to the next instruction.
    – Pay close attention to overridden conditions like "always face
forward" to prevent misinterpretation of implied directions.

**Example Processes**:

– Input: "Always face forward. Take 2 steps left. Take 4 steps
backward. Take 10 steps right."
  1. Start at (0, 0), facing north, with locked orientation forward
 (north).
  2. Move 2 steps left to (-2, 0).
  3. Move 4 steps backward to (-2, -4).
  4. Move 10 steps right to (8, -4). Final position: (8, -4).
  Output: `[{"label":"NO"}]`.

– Input: "Take 3 steps. Turn around. Take 3 steps."
  1. Start at (0, 0), facing north. Move 3 steps forward to (0, 3).
  2. Turn around to face south. Move 3 steps forward to (0, 0).
  3. Final position matches (0, 0).
  Output: `[{"label":"YES"}]`.

By prioritizing accurate parsing, intermediate validation, and
systematic processing of movements and orientation changes, ensure
consistent evaluations for all navigation tasks.

# Few-Shot Exemplars (from high-scoring failures)
[Example 1]
Input: If you follow these instructions, do you return to the
starting point? Always face forward. Take 2 steps left. Take 4 steps
```

```
 backward. Take 10 steps right. Take 2 steps left. Take 3 steps left
. Take 7 steps right.
Options:
- Yes
- No
Expected: NO

[Example 2]
Input: If you follow these instructions, do you return to the
starting point? Always face forward. Take 9 steps right. Take 6
steps right. Take 10 steps backward. Take 9 steps left. Take 4 steps
 left.
Options:
- Yes
- No
Expected: NO
```

**Prompt2**
```
You are an expert in navigation and spatial reasoning, tasked with
determining whether a series of movement instructions leads back to
the starting position (coordinate (0, 0) on a 2D Cartesian plane)...
```

**Prompt3**
```
You are a sophisticated navigation and spatial reasoning system
designed to determine whether a sequence of movement instructions
returns to the starting position, (0, 0), on a 2D Cartesian grid.
Your task is to ensure precise calculations and logical consistency
by strictly following the given instructions...
```

## D  ADDITIONAL RESULT

Here, we present the initial prompt and the ELPO-optimized prompt across different tasks.

**Initial prompt of the LIAR dataset**

```
## Task
Determine whether the Statement is a lie (Yes) or not (No) based on
the Context and other
information.
## Output format
Answer Yes or No as labels.
## Prediction
Text: {input}
Label:
```

**ELPO optimized prompt of the LIAR dataset**

**Prompt1:**
```
You are an expert in mathematics, logic, and navigational reasoning
tasked with rigorously determining whether a given Statement is true
 or false based solely on the provided Context and any explicitly
relevant, verifiable data. Follow this exact, methodical process:
carefully parse every word, focusing on precise interpretation of
negations, qualifiers, conditional phrases, and comparative or
superlative modifiers (e.g., "not," "only," "less than," "more than
," "ever," "always"); accurately interpret all quantitative
information including numbers, percentages, units, ratios, sequences
, temporal references, and ensure consistency with units and scales;
```

```
  thoroughly decompose complex or compound statements into smaller
components and verify each part individually; explicitly distinguish
 between facts directly supported by the Context or widely accepted
general knowledge and assumptions, opinions, or implicit claims,
avoiding any unsupported inferences; apply step-by-step logical
analysis, visualization, or spatial reasoning as appropriate to
validate relational, temporal, and logical claims; when information
is ambiguous, conflicting, or incomplete, prioritize the most direct
, explicit, and reliable evidence within the Context; double-check
all numerical calculations, logical deductions, and spatial
assessments before reaching a conclusion; remain vigilant against
common errors including misreading negations or qualifiers,
misinterpreting percentages or comparisons, overlooking units or
contextual cues, misclassifying assumptions as facts, and neglecting
 nested conditionals or compound relationships. Conclude with a
clear, concise final answer: output only "Yes" if the Statement is
verified as true by this rigorous analysis, or "No" if it is false.
Do not provide explanations or additional commentary.

# Few-Shot Exemplars (from high-scoring failures)
[Example 1]
Input: Says Amanda Fritz manages less than 5 percent of city
operations.
Expected: YES

[Example 2]
Input: Many state and federal agencies have such navigators involved
 in helping folks maneuver through the often complex processes
associated with filing benefits claims, for example -- even buying
health insurance.
Expected: YES

[Example 3]
Input: Says MAX carries 30 percent of evening rush-hour commuters
traveling from Downtown on the Sunset and Banfield freeways.
Expected: YES

[Example 4]
Input: The State of Texas is funding womens health services at
historically high levels; they just increased their level another 50
 million for the next two years.
Expected: YES

[Example 5]
Input: Ohio is not meeting its obligation to update voter
registrations when voters change their address with the BMV.
Expected: YES

[Example 6]
Input: Says he was the only Republican to vote against creating a
House panel to investigate Planned Parenthood.
Expected: YES

Prompt2:
## Task
As an expert in math and navigation reasoning, determine whether the
 given Statement is true or false based solely on the provided
Context and any explicitly verifiable, relevant information. Use
precise, step-by-step logical, numerical, temporal, and spatial
analysis without introducing unsupported assumptions.

## Guidelines
```

```
- Carefully parse every element of the Statement, paying special
attention to negations, conditionals, exclusive terms such as "only
," comparative phrases like "more than," "less than," and nuanced
modifiers.
- Break down complex or compound Statements into smaller parts and
verify each part independently against the Context.
- Cross-check all quantitative data, units, percentages, ratios,
dates, sequences, directions, and spatial relationships against the
Context, performing explicit calculations or logical reasoning as
needed.
- Distinguish rigorously between explicit facts and opinions,
assumptions, or implicit claims; rely solely on verifiable
information present in the Context or established, relevant general
knowledge.
- Avoid inferring or assuming information beyond the provided
Context unless it is logically necessary, explicitly justified, and
clearly documented in your reasoning.
- When confronted with ambiguity, contradiction, or incomplete
information, prioritize the most direct, explicit, and reliably
sourced evidence from the Context.
- Employ mental visualization, mapping, or systematic logical and
numerical checks to confirm temporal, spatial, conditional, and
comparative relationships.
- Double-check all calculations, logical deductions, qualifier
interpretations, and spatial or temporal conclusions before
finalizing your determination.
- Vigilantly avoid common errors such as ignoring negations,
misreading percentages or comparative data, conflating assumptions
with facts, misinterpreting spatial or logical relationships, or
overlooking key qualifiers.
- Maintain a rigorous, detailed, and cautious approach throughout
the analysis to ensure accuracy and reliability in your verification
.

## Output format
Respond only with a single word: Yes if the Statement is true
strictly based on the Context; otherwise, No.

## Prediction
Text: {input}
Label:

# Few-Shot Exemplars (from high-scoring failures)
[Example 1]
Input: This is the slowest job recovery since Hoover.
Expected: NO

[Example 2]
Input: The State of Texas is funding womens health services at
historically high levels; they just increased their level another 50
 million for the next two years.
Expected: YES

[Example 3]
Input: Oregonians have an amazing no-cost way to fight abortion with
 free political donations
Expected: YES

[Example 4]
Input: Our pension system is the only one in the country thats 100
percent funded.
Expected: YES
```

```
[Example 5]
Input: Says Rick Scott called education not a core function of the
state.
Expected: NO

[Example 6]
Input: When we took office, let me remind you, there was virtually
no international pressure on Iran.
Expected: NO

Prompt3:
## Task
As an expert in mathematics and spatial reasoning, determine whether
 the given Statement is true or false solely based on the supplied
Context and any directly relevant, verifiable information, using
thorough logical, numerical, and spatial analysis.

## Guidelines
- Perform a detailed and methodical review of all elements within
the Context before forming your conclusion.
- Accurately interpret every quantitative detail-including units,
ratios, percentages, sequences, directions, and spatial
relationships-without exception.
- Pay close attention to all negations, qualifiers, conditionals,
and implied meanings, carefully considering modifiers such as "not,"
 "only," "less than," "more than," and comparative terms.
- Distinguish clearly between facts, opinions, assumptions, and
implied statements; verify facts exclusively through explicit
contextual evidence or commonly accepted knowledge.
- Refrain from making assumptions beyond the provided information
unless they are strictly necessary, explicitly justified, and
clearly documented in your reasoning.
- In cases of ambiguity, contradiction, or incomplete data,
prioritize the most direct, explicit, and trustworthy information
available in the Context.
- Avoid common pitfalls: do not ignore negations or qualifiers,
misconstrue percentages or comparative data, or misunderstand
spatial, logical, or conditional relationships.
- Employ mental visualization, mapping, or stepwise logical checks
to confirm spatial and relational interpretations as needed.
- Verify all calculations, logical inferences, and spatial
evaluations carefully before delivering your final decision.
- Analyze the Statement meticulously, focusing on each word and
modifier in its context to ensure precise comprehension.

## Output format
Respond with a single word: Yes if the Statement is true; No if it
is false.

## Prediction
Text: {input}
Label:
```

**Initial prompt of the BBH-nevigate dataset**

```
# Role & Task
You are an expert in spatial navigation. Given a sequence of
navigation instructions, determine if the path returns to the
starting point.
```

```
Instructions include directions (north, south, east, west) and
actions (move forward, turn left, turn right).
Track the position and orientation carefully, considering each step
and turn. Output only "Yes" if the path returns to the starting
point, or "No" if it does not.

# Output Requirement
You must output the result in strict JSON format, with no additional
 text. The JSON should be an array containing a single object with
the "label" key.
- If the path returns to the starting point, output: [{"label":"YES
"}]
- If the path does NOT return to the starting point, output: [{"
label":"NO"}]
- Any output that deviates from this format (such as plain text,
missing brackets, or other content) will be considered invalid.

# Example
- Input: "Always face forward. Take 1 step backward. Take 4 steps
left. Take 4 steps left."
- Output: [{"label":"NO"}]
```

## ELPO optimized prompt of the BBH-nevigate dataset

**Prompt1**
```
You are an expert in spatial navigation path - return - to - start -
 point determination. Given a series of navigation instructions (
which consist of directions like north, south, east, west and
actions like move forward, turn left, turn right), figure out
whether the path goes back to the starting point.
- **Orientation Tracking**: Start by initializing the orientation (
assume facing north at the start). Maintain a turn - count (mod 4)
and orientation mapping (turn_count: 0 to North, 1 to West, 2 to
South, 3 to East). For each turn (left/right), update the
orientation using the cumulative 90 - degree turn rule. After every
4 left/right turns, reset the orientation. Keep a running count of
cumulative turn angles.
- **Movement Tracking**: For each forward/backward step, update the
position in the current orientation's forward/backward axis (e.g.,
if facing north, forward is +y, backward is -y in a Cartesian - like
 coordinate system. If facing east, forward is +x, backward is -x.
If facing south, forward is -y, backward is +y. If facing west,
forward is -x, backward is +x). When there is a left/right step (
which changes the direction of movement), first update the
orientation as per the rules and then update the position. Track
movements explicitly by writing down the movement in each axis (x
and y) for each step (e.g., [orientation, x_change, y_change]).
- **Sum Calculation**: Maintain separate sums for the x (east - west
) and y (north - south) axes. After processing all the steps in the
instruction, check if both sums are zero. Use a step - by - step
table for x and y sums and double - check arithmetic (e.g., - 5+3 =
- 2, not + 2). Use intermediate checks (e.g., after every 5 steps).
- **Edge Case Handling**: Test edge cases like 4 consecutive turns (
left or right) to ensure orientation reset. For ambiguous inputs (e.
g., "Turn around" = 2 left/right turns), convert to standard turns.
For movement without direction (e.g., "Take N steps"), assume
forward unless context (like prior "Turn around") implies backward.
Pre - process input: replace "Turn around" with "Turn left Turn left
" or "Turn right Turn right". For "Take N steps", default to "Take N
 steps forward" (override if "Turn around" precedes).
```

- **Error Prevention**: Avoid orientation miscalculation (especially orientation reset after 4 turns), movement axis error (correctly map movement to current orientation's axis), sum calculation arithmetic mistake, and edge case neglect. Use a systematic approach (like writing down orientation and movement for each step) to avoid confusion. Double – check all calculations, especially for edge cases. Pay close attention to the order of operations (e.g., update orientation first when there is a turn before updating movement). When handling input instructions, parse them carefully to ensure all steps (turns and movements) are correctly identified and processed. When dealing with movement steps that have no direction specified (e.g., "Take 10 steps" without specifying forward or backward), assume forward movement unless context suggests otherwise. But also be aware of cases where "turn around" followed by "Take steps" implies backward movement.

# Few-Shot Exemplars (from high-scoring failures)
[Example 1]
Input: If you follow these instructions, do you return to the starting point? Always face forward. Take 2 steps forward. Take 2 steps backward. Take 4 steps right. Take 7 steps right.
Options:
- Yes
- No
Expected: NO

**Prompt2**:
You are an expert in spatial navigation path – return – to – start – point determination. Given a series of navigation instructions ( which consist of directions like north, south, east, west and actions like move forward, turn left, turn right), figure out whether the path goes back to the starting point.
- **Orientation Tracking**: Start by initializing the orientation ( assume facing north at the start). Maintain a turn – count (mod 4) and orientation mapping (turn_count: 0 to North, 1 to West, 2 to South, 3 to East). For each turn (left/right), update the orientation using the cumulative 90 – degree turn rule. After every 4 left/right turns, reset the orientation. Keep a running count of cumulative turn angles. Replace "Turn around" with "Turn left Turn left" or "Turn right Turn right" (whichever is appropriate).
- **Movement Tracking**: For each movement (e.g., "Take N steps forward/backward"):
    – If there was a turn before the movement, update orientation first.
    – Map movement to current orientation's axis:
        – North: forward = +y, backward = −y
        – East: forward = +x, backward = −x
        – South: forward = −y, backward = +y
        – West: forward = −x, backward = +x
    – Record x_change and y_change for each step (e.g., [orientation, x_change, y_change]).
    – For movement without direction (e.g., "Take N steps"):
        – Assume forward unless "Turn around" precedes (then assume backward).
- **Sum Calculation**: Maintain separate sums for x and y axes. After each step, update the sums (e.g., x_sum += x_change, y_sum += y_change). Use intermediate checks (e.g., after every 5 steps) to verify sums. Double – check arithmetic (e.g., −5 + 3 = −2, not +2).
- **Edge Case Handling**: Test edge cases like 4 consecutive turns ( left or right) to ensure orientation reset. For ambiguous inputs (e. g., "Turn around"), convert to standard turns (2 left/right turns). For movement without direction, use the default (forward) with context override (if "Turn around" precedes, use backward).

```
- **Error Prevention**: Avoid orientation miscalculation (especially
  orientation reset after 4 turns), movement axis error (correctly
map movement to current orientation's axis), sum calculation
arithmetic mistake, and edge case neglect. Use a systematic approach
  (like writing down orientation and movement for each step in a
table: Step \ Instruction \ Orientation \ x_change \ y_change) to
avoid confusion. Double - check all calculations, especially for
edge cases. Pay close attention to the order of operations (e.g.,
update orientation first when there is a turn before updating
movement). When handling input instructions, parse them carefully to
 ensure all steps (turns and movements) are correctly identified and
 processed. Additionally, always use a step - by - step table (as
mentioned in the error prevention section) to avoid confusion. After
 calculating x_sum and y_sum, double - check the arithmetic and
ensure that orientation was correctly updated before each movement.
Familiarize yourself with edge cases (4 turns, ambiguous
instructions) through practice problems. Use the following
additional guidelines:
   - **Orientation Tracking**: Always start with initial orientation
   (north) and turn - count (0). For each turn (left/right), increment
/decrement turn - count (mod 4). Double - check orientation mapping
(0 to North, 1 to West, 2 to South, 3 to East). When 4 turns (left
or right) occur consecutively, reset turn - count to 0 and
orientation to North. Replace "Turn around" with 2 left/right turns
(whichever is appropriate) immediately.
   - **Movement Tracking**: If a turn precedes a movement, update
orientation first. Use the correct axis mapping: North (forward = +y
, backward = -y), East (forward = +x, backward = -x), South (forward
 = -y, backward = +y), West (forward = -x, backward = +x). For
movement without direction (e.g., "Take N steps"), assume forward
unless "Turn around" precedes (then assume backward). Record [
orientation, x_change, y_change] for each step in a table (Step \
Instruction \ Orientation \ x_change \ y_change).
   - **Sum Calculation**: Maintain separate x_sum and y_sum. After
each step, update the sums (x_sum += x_change, y_sum += y_change).
Use intermediate checks (e.g., after every 5 steps) to verify sums.
Double - check all arithmetic operations (e.g., - 5+3=-2, not +2).
   - **Edge Case Handling**: Test 4 consecutive turns (left or right
) in practice problems to ensure orientation reset. For ambiguous "
Turn around", convert to standard turns (2 left/right turns) as per
system prompt. For movement without direction, use default (forward)
 with context override (if "Turn around" precedes, use backward).
   - **General Advice**: Use a systematic step - by - step table (as
 in the error prevention section of the system prompt) for every
problem. This helps in visualizing orientation and movement changes
clearly. Double - check all calculations (orientation update,
movement - axis mapping, sum arithmetic) at each step. Familiarize
yourself with common edge cases (4 turns, ambiguous instructions)
through continuous practice. When in doubt about an instruction (e.g
., movement direction ambiguity), refer back to the rules in the
system prompt (e.g., default forward with "Turn around" context
override for backward).

# Few-Shot Exemplars (from high-scoring failures)
[Example 1]
Input: If you follow these instructions, do you return to the
starting point? Always face forward. Take 6 steps backward. Take 8
steps left. Take 3 steps right. Take 7 steps forward. Take 3 steps
right. Take 9 steps right. Take 1 step backward. Take 7 steps left.
Options:
- Yes
- No
Expected: YES
```

```
[Example 2]
Input: If you follow these instructions, do you return to the
starting point? Always face forward. Take 8 steps forward. Take 9
steps right. Take 2 steps forward. Take 10 steps backward. Take 9
steps left.
Options:
- Yes
- No
Expected: YES

[Example 3]
Input: If you follow these instructions, do you return to the
starting point? Always face forward. Take 9 steps left. Take 10
steps forward. Take 9 steps right. Take 10 steps backward.
Options:
- Yes
- No
Expected: YES
```

**Initial prompt of the ETHOS dataset**

```
## Task
Is the following text hate speech?
## Output format
Answer Yes or No as labels.
## Prediction
Text: {input}
Label:
```

**ELPO optimized prompt of the ETHOS dataset**

**Prompt1**:
You are an expert in identifying hate speech in text. Hate speech is
 language that explicitly or implicitly targets a protected group-
defined by characteristics including race, ethnicity, religion,
gender, sexual orientation, disability, mental or physical health
status, or other identities-with clear hostility, hateful intent, or
 calls for hatred, discrimination, violence, or dehumanization
against that group.

To determine whether the given text constitutes hate speech, follow
these precise guidelines:

1. Confirm that the text targets a protected individual or group
with explicit or implicit hostility, hateful intent, or advocacy of
hatred, discrimination, violence, or dehumanization. The hostility
must be connected to the group's identity; general criticism,
negative opinions, or frustration without group-based hostility do
not qualify as hate speech.

2. Recognize that hate speech includes not only overt hateful
statements but also subtle, coded, metaphorical, sarcastic, or
indirect language that clearly conveys hostility and intent to
demean, devalue, or incite harm toward a protected group.

3. Carefully distinguish between insults or negative remarks about
individuals or behaviors and hate speech. Individual insults count
as hate speech only if they target the person explicitly because of

31
```

```
their protected group identity or as a representative symbol of that
 group.

4. Apply a comprehensive and inclusive list of protected groups,
explicitly including mental and physical health conditions as
protected statuses. Statements attacking or dehumanizing people on
the basis of mental or physical illness, or advocating harm or
discrimination toward them, constitute hate speech.

5. Analyze the context, rhetorical devices, and implied intention
behind language to identify if the text promotes hatred,
discrimination, violence, or dehumanization against a protected
group.

6. Require clear, convincing, and direct or indirect evidence of
hateful targeting and harmful intent before labeling text as hate
speech. Avoid false positives by responding 'No' when evidence is
ambiguous, lacks clear hostility toward a protected group, or only
expresses criticism or negative opinion without hateful intent.

7. When uncertain or lacking sufficient evidence of hateful intent
toward a protected group, always respond 'No.' Respond only with '
Yes' if the text clearly meets all these criteria.

Respond only with 'Yes' if the text unequivocally constitutes hate
speech as defined above; otherwise, respond with 'No.'

Text: {input}
Label:

# Few-Shot Exemplars (from high-scoring failures)
[Example 1]
Input: To say that black lives matter, is not racist statement. But
the B.L.M. organization itself is racist group. If they could just
quit hating white people and turn that energy towards their own
community, the biggest group of black oppressors there is, they
might make a difference.
Expected: NO

[Example 2]
Input: Do you know why he is smiling because there is no 'excretion
law' in New Zealand! The max sentence he will receive from a judge
is no more than 27 years in prison! Is this justice? Or because
Muslims lives don't matter!??? :(((
Expected: NO

[Example 3]
Input: I'm very sad about this fucking situation with all this
muslims
Expected: NO

[Example 4]
Input: all mental illnesses are awful and must be treated
Expected: YES

Prompt2:
You are an expert in identifying hate speech in text. Hate speech is
 language that explicitly or implicitly targets a protected group-
defined by characteristics including race, ethnicity, religion,
gender, sexual orientation, disability, mental or physical health
status, or other identities-with the intent to promote hatred,
discrimination, violence, dehumanization, or to incite such actions.
```

```
To determine whether the given text constitutes hate speech, follow
these precise steps:

1. Identify whether the text targets a protected group or
individuals specifically identified with a protected group with
hostility, hatred, harmful intent, or calls for harm, including
explicit or implicit hateful language conveyed through direct,
subtle, coded, or indirect means.

2. Recognize that hate speech may include aggressive insults,
threats, dehumanization, calls for violence, or derogatory language
linked to protected group identities, even if veiled by sarcasm,
metaphor, or rhetorical devices.

3. Distinguish hate speech from language expressing general anger,
frustration, criticism, or insults that do not specifically target
protected groups or advocate harm against them; profanity or insults
 alone do not constitute hate speech without targeted hateful intent
.

4. Carefully analyze context, underlying intent, rhetorical
structures, and the relationship between the language and protected
group identities to detect advocacy, promotion, or encouragement of
hatred, discrimination, violence, or dehumanization.

5. Include all protected groups comprehensively, explicitly
considering commonly overlooked identities such as mental and
physical health conditions, religion, ethnicity, and political or
ideological groups when they are targeted with hateful intent.

6. Treat insults or threats aimed solely at individuals as hate
speech only if they reflect hostility toward the individual's
protected group identity or are directed at individuals as
representatives of a protected group.

7. Require clear, convincing evidence of hateful targeting and
harmful intent before labeling text as hate speech, avoiding false
positives by withholding judgment when such evidence is ambiguous or
 lacking.

8. When uncertain, err on the side of caution by responding 'No';
respond only with 'Yes' if the text clearly meets all criteria of
hate speech as defined above.

Respond only with 'Yes' if the text clearly meets these criteria;
otherwise, respond with 'No.'

Text: {input}
Label:

# Few-Shot Exemplars (from high-scoring failures)
[Example 1]
Input: That girl still need a good ass whooping, stupid ass bitch!!
Expected: YES

[Example 2]
Input: This priest is being hated even by his stupid god.
Expected: YES

[Example 3]
Input: If I had to meet one of the apostles I would fuck them for
not killing him
Expected: YES
```

```
[Example 4]
Input: Ignorant feminist, she got roasted and still doesn'
understand what equality of outcome means even when Jordan explained
 data to her. What's this means she's ignorant and emotionally
biased.
Expected: YES

[Example 5]
Input: Taliban and all people with this thought must hanged
Expected: YES
```

**Prompt3**:
```
You are tasked with determining whether a given text constitutes
hate speech. Hate speech is defined as language that explicitly or
implicitly targets a protected group-such as those defined by race,
ethnicity, religion, gender, sexual orientation, disability, mental
or physical health status, or other identities-with the intent to
promote hatred, discrimination, violence, dehumanization, or to
incite such actions.

To accurately assess if the text is hate speech:
1. Identify whether the text targets a protected group in a hateful
or harmful way, either explicitly or through subtle, coded, or
indirect language.
2. Differentiate hate speech from negative or emotionally charged
expressions like anger, frustration, criticism, sarcasm, profanity,
or insults that do not promote hatred or harm against protected
groups. Strong emotions or controversial opinions alone do not
constitute hate speech.
3. Consider the context, rhetorical devices, and underlying intent
behind the language to detect if there is an advocacy or promotion
of hatred, discrimination, or violence toward the protected group.
4. Include all protected groups comprehensively, paying particular
attention to those related to mental or physical health conditions
as these are often overlooked.
5. Avoid labeling statements expressing sorrow, anger, or
frustration without hateful targeting as hate speech.
6. Use a careful, methodical approach, applying hate speech
definitions consistently, and strive to minimize both false
positives (wrongly labeling non-hate speech as hate speech) and
false negatives (missing actual hate speech).
7. When uncertain, prioritize accuracy by balancing caution and
rigor, seeking contextual clues before labeling.

Respond only with 'Yes' if the text meets the criteria for hate
speech; otherwise, respond with 'No.'

Text: {input}
Label:

# Few-Shot Exemplars (from high-scoring failures)
[Example 1]
Input: To say that black lives matter, is not racist statement. But
the B.L.M. organization itself is racist group. If they could just
quit hating white people and turn that energy towards their own
community, the biggest group of black oppressors there is, they
might make a difference.
Expected: NO

[Example 2]
Input: Can't tell if you're a pedophile victim or a pedophile.
Expected: NO
```

```
[Example 3]
Input: Do you know why he is smiling because there is no 'excretion
law' in New Zealand! The max sentence he will receive from a judge
is no more than 27 years in prison! Is this justice? Or because
Muslims lives don't matter!??? :(((
Expected: NO

[Example 4]
Input: all mental illnesses are awful and must be treated
Expected: YES

[Example 5]
Input: or when the man walked up to the little girl with the gay
pride flag i wanted to jump into the video and punch him but ya know
 thats assault
Expected: YES
```

## Initial prompt of the ArSarcasm dataset

```
## Task
Is this tweet sarcastic?
## Output format
Answer Yes or No as labels.
## Prediction
Text: {input}
Label:
```

## ELPO optimized prompt of the ArSarcasm dataset

**Prompt1**:
```
## Task
Determine whether the following tweet is sarcastic. Sarcasm often
involves saying the opposite of what is meant, using irony,
exaggeration, or humor to convey criticism or mockery.

## Instructions
1. Carefully analyze the tone, context, and implicit meaning of the
tweet.
2. Consider whether the statement uses irony, ridicule, or
exaggeration to convey a message contrary to the literal words.
3. Tweets may include cultural, political, or social references;
account for these nuances.
4. If unsure, apply a step-by-step reasoning process (Chain of
Thought) to reflect on the indicators of sarcasm before deciding.
5. Do NOT label as sarcastic if the statement is straightforward or
literal without hints of irony or mockery.
6. Your final answer must be either "Yes" (sarcastic) or "No" (not
sarcastic), with no additional text.

## Prediction
Text: {input}
Thought process: [Step-by-step reasoning about tone, context, irony,
 exaggeration, cultural references, and intent]
Label:
```
**Prompt2**:
```
You are an expert in detecting sarcasm in short Arabic tweets. For
each tweet, carefully analyze subtle linguistic cues, tone, cultural
 context, and the use of irony, exaggeration, or contradiction
```

```
between the literal content and the intended meaning. Pay close
attention to indirect expressions, humor, hashtags, emojis,
interjections, and any mockery or criticism disguised as praise or
neutral statements. Consider cultural references and context beyond
just the words to determine if the tweet's message contradicts
reality or conveys criticism through sarcasm. For each input, first
reflect on potential misinterpretations by identifying missed irony,
 overlooked mockery, or ignored cultural context. Then answer with
Yes if the tweet is sarcastic, otherwise No.

Text: {input}
Label:

Prompt3:
## Task
Determine whether the following tweet is sarcastic. Sarcasm
typically involves expressing the opposite of the literal meaning,
often using irony, exaggeration, mockery, or humor to convey
criticism or a hidden message.

## Instructions
1. Carefully analyze the tweet's literal meaning first before
considering sarcasm.
2. Identify clear, explicit or strongly implied indicators of
sarcasm such as irony, ridicule, exaggeration, contradictory
statements, or mockery.
3. Consider cultural, political, social, and linguistic
referencesincluding emojis, hashtags, and idiomatic expressionsbut
treat these only as supporting evidence, never as definitive proof
of sarcasm.
4. When tone or context is subtle or ambiguous, apply a deliberate
step-by-step Chain of Thought reasoning process: weigh all
linguistic and contextual clues objectively, verifying if the
message contradicts its literal meaning or contains mockery or irony
.
5. Do not label a tweet sarcastic if it is straightforward, literal,
 serious, or lacks clear cues of mockery, humor, irony, or
contradiction.
6. Avoid overinterpreting emojis, hashtags, or cultural references
without supporting textual evidence of sarcasm.
7. Your final answer must be exactly "Yes" if sarcastic or "No" if
not, with no additional explanation or text.

## Prediction
Text: {input}
Thought process: [Detailed, stepwise analysis of literal meaning,
tone, irony, exaggeration, context, cultural and linguistic cues,
verifying contradictions or mockery before concluding sarcasm]
Label:
```

**Initial prompt of the WSC dataset**

```
## Task
Solve the problem.
## Prediction
Text: {input}
Label:
```

**Initial prompt of the WSC dataset**

**Prompt1**:
You are an expert in pronoun resolution and coreference
understanding within short text passages that require nuanced
semantic and logical reasoning. Given a passage and two options
labeled A and B, your task is to determine the noun phrase that the
pronoun logically and semantically refers to, based on a thorough
analysis of the passages full context.

Guidelines:
1. Carefully read the entire passage to fully grasp the situation
and the roles of all entities mentioned.
2. Identify all plausible noun phrase candidates that the pronoun
could refer to; do not assume the closest noun phrase is correct.
3. For each candidate, evaluate semantic compatibility, actions
described, properties, and the logical coherence of the pronouns
reference within the passage.
4. Prioritize logical and semantic fit over proximity or surface
cues.
5. Eliminate candidates that conflict with the passages meaning,
actions, or described properties.
6. Select the antecedent that best maintains overall coherence,
logical consistency, and natural interpretation of the passage.

Submit only the letter of the correct answer.

Example:
Text: "I couldn't find a spoon, so I tried using a pen to stir my
coffee. But that turned out to be a bad idea, because it got full of
 ink. What does the pronoun 'it' refer to?" (A) The pen (B) The
coffee
Label: B

Prediction:
Text: {input}
Label:

# Few-Shot Exemplars (from high-scoring failures)
[Example 1]
Input: "I couldn't find a spoon, so I tried using a pen to stir my
coffee. But that turned out to be a bad idea, because it got full of
 ink. What does the pronoun 'it' refer to?" (A) The pen (B) The
coffee
Expected: B

**Prompt2**:
## Task
Solve the problem by choosing the appropriate option, either A or B,
 and submit only the letter of your chosen answer.
## Example
Text: "Steve follows Fred's example in everything. He admires him
hugely. What does the pronoun 'He' refer to?" (A) Steve (B) Fred
Label: A
## Prediction
Text: {input}
Label:
**Prompt3**:
You are an expert in resolving pronoun ambiguity in complex
sentences containing multiple clauses and potential antecedents. For
 each input sentence, determine whether option A or B correctly
identifies the pronouns true referent. Provide your answer as a

```
single letter, preceded by a thorough, step-by-step analysis as
detailed below.

Follow this comprehensive procedure before selecting your answer:

1. Exhaustively identify every plausible antecedent from all
clausesmain, subordinate, embedded, and causally linkedwithout
omitting or assuming candidates based on proximity, salience, or
default heuristics.

2. For each candidate antecedent, rigorously verify grammatical
agreement in person, number, gender, and syntactic compatibility,
carefully analyzing the sentences structure.

3. Fully parse the sentences syntax to delineate clause hierarchies
and establish the precise grammatical role (e.g., subject, object,
possessor) of each candidate within all relevant clauses.

4. Integrate deep semantic and contextual reasoning, assessing
coherence, causal relationships, and real-world plausibility for
each candidate as the pronouns referent, considering who logically
can perform or experience the described action.

5. Avoid premature exclusion of any candidates; only eliminate
antecedents after thorough syntactic and semantic evaluation.

6. If ambiguity persists after the initial pass, systematically
repeat all steps, ensuring no candidates have been overlooked or
incorrectly rejected and that both syntactic and semantic analyses
are fully complete.

Do not rely on shortcuts such as defaulting to the nearest noun,
using gender cues alone, or making unsubstantiated assumptions.

Format your response exactly as follows:

Text: {input sentence with options}
Label: {A or B}

# Few-Shot Exemplars (from high-scoring failures)
[Example 1]
Input: "Tom threw his schoolbag down to Ray after he reached the
bottom of the stairs. What does the pronoun 'he' refer to?" (A) Tom
(B) Ray
Expected: B

[Example 2]
Input: "John couldn't see the stage with Billy in front of him
because he is so short. What does the pronoun 'he' refer to?" (A)
John (B) Billy
Expected: A

[Example 3]
Input: "Madonna fired her trainer because she couldn't stand her
boyfriend. What does the pronoun 'her' refer to?" (A) Madonna (B)
The trainer
Expected: B
```

### Initial prompt of the GSM8K dataset

```
## Task
Solve the math problem.
## Prediction
Input: {input}
Expected:
```

### Initial prompt of the GSM8K dataset

**Prompt1**:
You are an expert in solving complex multi-step math word problems
involving quantities, totals, leftovers, and transactional
relationships such as items ordered, sold, leftover, or remaining.
For each problem:

1. Carefully read the entire problem and identify all given
quantities, explicitly extracting each with its units and clear
labels. Define distinct variables for every relevant quantity
involved.

2. Pay close attention to relational language and key terms like
leftover, remaining, total, sold, ordered, and similar. Precisely
translate these into explicit mathematical relationships for example
, interpret "leftover" as items remaining after subtraction (
leftover = ordered  sold).

3. Write down all equations representing these relationships before
performing any calculations, clearly linking totals to sums or
differences of parts. Explicitly state how quantities combine,
increase, decrease, or remain consistent.

4. Systematically solve the problem step-by-step: carry out
arithmetic operations carefully, double-check all calculations
immediately after each step, and verify that the operations
correctly reflect the relationships identified.

5. After computing intermediate and final results, verify their
numerical correctness, logical coherence, and contextual
consistencyincluding proper units and labelsand ensure all
quantities sum or balance as indicated by the problem. If
inconsistencies appear, revisit and correct prior steps before
proceeding.

6. Use estimation and reasonableness checks throughout to confirm
answers are plausible within the problems context and scale.

7. Present only the final numeric answer exactly as requested,
including units if specified, without explanation, intermediate
steps, or commentary.

Apply this methodical approach to all problems to accurately track
quantities, totals, leftovers, and related multi-step arithmetic
reasoning.

# Few-Shot Exemplars (from high-scoring failures)
[Example 1]
Input: Barney's grocery store sold out all of its items at the
beginning of the pandemic, so they ordered extra items to restock
the shelves. However, they ended up ordering far too much and have
to keep the leftover items in the storeroom. If they ordered 4458
```

```
items, sold another 1561 items that day, and have 575 items in the
storeroom, how many items do they have left in the whole store?
Expected: 3,472

Prompt2:
You are proficient at tackling complex multi-step math word problems
 covering arithmetic, ratios, proportions, percentages, geometry,
and fundamental algebra. For each given problem, adhere to this
meticulous procedure:

1. Thoroughly read the problem and explicitly identify every
numerical value presented, including their units and relevant
context (such as totals, parts, rates, or specific conditions).

2. Clearly assign variables and extract all expressed relationships,
 ratios, proportions, and conditions from the problem; accurately
convert these into precise mathematical equations or expressions
without making any assumptions beyond the provided information.

3. Break the problem down into well-defined, logical, and sequential
 steps. Fully solve and confirm the correctness of each step before
proceedingdo not omit any intermediate calculations or reasoning.

4. Ensure units are consistently and correctly applied throughout
all computations, converting units beforehand when necessary.

5. Process percentages, ratios, and fractions with careful attention
 to their accurate contextual meanings.

6. After completing each intermediate calculation, review its
validity, consistency, and plausibility within the problems scenario
.

7. Once all steps are complete and verified, thoroughly re-examine
the entire solution to ensure it fully satisfies the questions
demands and constraints, double-checking all interpretations,
equation setups, and mathematical procedures.

8. At the conclusion, provide only the precise final numerical
answer requested, including units if specified, with no explanations
, intermediate details, or commentary.
Prompt3:
## Task
Solve the math problem and provide only the final answer without any
 extra explanations or comments.
# Few-Shot Exemplars
Input: Ryan is considering buying a new multivitamin brand. Each
pill contains 50 mg of Vitamin A. The recommended daily intake of
Vitamin A is 200 mg. How many pills does Ryan need to consume in a
week to meet the recommended amount?
Expected: 28
## Prediction
Input: {input}
Expected:
```

