# OpenReview forum: "ELPO: Ensemble Learning Based Prompt Optimization for Large Language Models"
_ICLR.cc/2026/Conference — ICLR 2026 Conference Withdrawn Submission_

### Official Review · Reviewer_s7p7 · 2025-10-21

**Soundness:** 3
**Presentation:** 2
**Contribution:** 3
**Rating:** 4
**Confidence:** 4

**Summary:**

This paper introduces (ELPO, a novel framework that enhances the effectiveness of LLMs through automatic prompt optimization. Unlike existing approaches that depend on a single generation or search strategy, ELPO adopts an ensemble learning paradigm that integrates multiple search methods and shared generation strategies under a voting mechanism, enabling more accurate and robust prompt discovery. Furthermore, ELPO proposes efficient algorithms for prompt generation and search, improving both optimization stability and adaptability across diverse tasks. Extensive experiments show that ELPO consistently outperforms state-of-the-art prompt optimization methods, demonstrating its strong generalization ability and superior performance on various benchmarks.

**Strengths:**

The main strength of this paper is its significant performance improvement over existing prompt optimization methods. The proposed ELPO framework achieves consistently higher results across multiple benchmarks, clearly demonstrating its effectiveness in optimizing prompts for Large Language Models. By leveraging ensemble learning to combine diverse search and generation strategies, ELPO enhances the quality of discovered prompts and achieves state-of-the-art performance in various evaluation settings. The results convincingly show that the ensemble-based design leads to more accurate and reliable optimization outcomes, establishing ELPO as a strong and effective approach within the field of automatic prompt optimization.

**Weaknesses:**

1. Unclear technical novelty and framing. The paper does not clearly articulate which components are genuinely novel versus straightforward extensions. While results suggest that incorporating diverse generators drives most of the gains, the manuscript does not clearly explain how these generators are integrated. Is there any critical design?

2. Unsupported claims about Bayesian Search + MAB. The introduction claims the method is the first to combine Bayesian optimization with MAB, but there is no targeted ablation or analysis isolating this design choice.

3. Hard-Case Tracking lacks differentiation. The paper states it “creatively” proposes Hard-Case Tracking focusing on recurrent error samples, but leveraging error cases for iterative refinement is common. It remains unclear how ELPO’s variant differs from prior methods.

4. Incomplete ablations on key design choices. ELPO contains several moving parts (ensemble of generators, voting, search components...), yet the paper does not rigorously demonstrate the contribution of each (e.g. search).

5. No evaluation of efficiency. The introduction claims “substantial” efficiency improvements, but there are no metrics or token-cost analyses to substantiate this.

6. Limited model coverage undermines robustness claims. Experiments are run only on GPT-4o, which makes it hard to assess robustness or generalization across base models. Replicate on multiple families (e.g., Claude, Gemini, Qwen, Llama) to demonstrate model-agnostic benefits.

**Questions:**

See the weaknesses.

---

### Official Review · Reviewer_gCQa · 2025-10-30

**Soundness:** 2
**Presentation:** 3
**Contribution:** 2
**Rating:** 2
**Confidence:** 4

**Summary:**

While Large Language Models (LLMs) are powerful, their performance hinges on expertly designed prompts. Since manual prompt engineering is labor-intensive, and Automatic Prompt Optimization (APO) aims to optimize such scenarios. Current APO methods, though somewhat effective, are often limited to a single technique, which struggles with complex tasks. This paper proposes a framework called Ensemble Learning-based Prompt Optimization (ELPO). ELPO's key innovations are: 1) Hard-Case Tracking to learn from repeated errors, 2) a high-dimensional Bayesian search for efficient optimization, and 3) ensemble voting that leverages a diverse set of high-performing prompts.

**Strengths:**

* The method is clear to understand and this paper is well-written.
* The proposed ELPO achieves remarkable scores on several datasets compared with baselines.

**Weaknesses:**

* The contributions are somehow incremental, ensembling previous prompt optimization methods with careful and further design in three phases: 1) prompt generation by the following methods: a) bad case reflection[1], b) evoprompt[2], c) OPRO; 2) search phrase: a) Bayesian, b) UCB[1], c) population[2,4]; 3) Voting: intuitive ensembling voting and aggregating.
* The method is not very novel, just aggregation of previous methods.
* The experimental setting including parameters are not given. How many steps does each method cost in the table 1?

# References
[1] Pryzant, Reid, et al. "Automatic Prompt Optimization with" Gradient Descent" and Beam Search." The 2023 Conference on Empirical Methods in Natural Language Processing.
[2] Guo, Qingyan, et al. "Connecting Large Language Models with Evolutionary Algorithms Yields Powerful Prompt Optimizers." The Twelfth International Conference on Learning Representations.
[3] Yang, Chengrun, et al. "Large language models as optimizers." The Twelfth International Conference on Learning Representations. 2023.
[4] Zhou, Yongchao, et al. "Large language models are human-level prompt engineers." The eleventh international conference on learning representations. 2022.

**Questions:**

* When it comes to ensemble, trade-offs between overhead and performances are very important. Could the authors give more in-depth cost analysis, both for time and API cost?
* See weakness part.

---

### Official Review · Reviewer_LBgm · 2025-10-30

**Soundness:** 3
**Presentation:** 3
**Contribution:** 3
**Rating:** 4
**Confidence:** 3

**Summary:**

This paper introduces ELPO, a new framework for Automatic Prompt Optimization (APO). The authors argue that existing APO methods are often limited because they rely on a single generation strategy or optimization algorithm, which can struggle with complex tasks. To address this, ELPO adopts an ensemble learning approach across the entire optimization pipeline: It first creates a diverse pool of candidate prompts by using three parallel generation strategies: Bad-Case Reflection, Evolutionary Reflection, and Hard-Case Tracking. To avoid the high cost of evaluating every candidate prompt, ELPO uses two parallel search algorithms to select the most promising ones. They believe this kind of combination can complement each method. Finally, instead of selecting a single best prompt, ELPO aggregates predictions from multiple high-performing, diverse prompts. It uses a weighted voting mechanism, where weights are optimized to maximize the macro F1-score, to produce the final output.

The authors evaluate ELPO on six datasets (including LIAR, BBH, and GSM8K)  and show that it consistently outperforms state-of-the-art baselines like APE, ProTeGi, Promptbreeder, and EvoPrompt etc. Ablation studies confirm the positive contribution of the diverse generators and the ensemble voting strategy.

**Strengths:**

1.  The central idea of applying ensemble learning principles to the entire APO pipeline (generation, search, and final prediction) is a good attempt for research. It directly addresses a well-known limitation of single-algorithm optimizers.
2.  The method demonstrates significant and consistent performance gains over a wide range of strong, recent baselines across six diverse tasks. The improvements on challenging datasets like LIAR and BBH-navigate are particularly impressive.
3.  The proposed Hard-Case Tracking is an interesting method that addresses a key weakness in feedback-based optimizers by moving from a myopic, single-prompt error view to a global, population-wide "hard case" analysis.
4.  The authors provide clear ablation studies in Table 2 and Table 3 that effectively validate the contributions of the multi-generator framework and the weighted voting strategy, respectively.

**Weaknesses:**

1.  The framework's primary weakness is its significant complexity and likely computational cost. ELPO runs three generation algorithms, two search algorithms, and an additional weight optimization step. This is almost certainly more expensive (in terms of LLM calls to the optimizer) than the single-algorithm baselines. The paper claims "efficiency"  and "conserves resources", but this is only relative to a naïve evaluation of all candidates, not relative to the baselines. The lack of a cost-performance comparison makes it difficult to assess if ELPO is truly more efficient or just finds better prompts through a much more expensive search.
2.  The Ensemble Voting mechanism requires running M different prompts through the task LLM for every single query at inference time. This multiplies inference cost and latency by M. This is a major practical drawback that seems to conflict with the paper's motivation of solving a "core bottleneck for practical application". This trade-off is not discussed.
3.  It is unclear how the two parallel search methods (Bayesian and MAB) interact. Are their results pooled? Do they select prompts in sequence? The pipeline diagram and text do not sufficiently detail how this dual-search mechanism is unified to produce the set of prompts for evaluation.
4. Another concern here is whether we still need to search for more robust and effective prompting methods nowadays. Since the power of LLM is now quite strong, and can tolerate even typos or simple logical problems.

**Questions:**

1.  Could the authors provide a comparative analysis of the total computational cost (e.g., total number of optimizer LLM calls) required for ELPO to reach its reported performance versus the baseline methods? A plot of final performance vs. LLM calls would be very insightful.
2.  Following on the high inference cost of ensemble voting: How does the performance of the single best prompt found by ELPO (before the voting step) compare to the final ensemble result and the baselines? This would help disentangle the gains from the superior search process versus the expensive inference-time ensemble.

---

### Official Review · Reviewer_ML5W · 2025-11-01

**Soundness:** 2
**Presentation:** 2
**Contribution:** 2
**Rating:** 4
**Confidence:** 3

**Summary:**

This paper proposes ELPO (Ensemble Learning based Prompt Optimization), a framework for automatic prompt optimization that combines multiple generation strategies and search methods with ensemble voting. The approach introduces three prompt generation methods (Bad-Case Reflection, Evolutionary Reflection, and Hard-Case Tracking), two efficient search strategies (Bayesian Search and Multi-Armed Bandit Search), and a weighted ensemble voting mechanism. Experiments on six datasets (LIAR, BBH-navigate, ETHOS, ArSarcasm, WSC, GSM8K) demonstrate improvements over existing methods like APE, ProTeGi, OPRO, PromptBreeder, and EvoPrompt.

**Strengths:**

- ELPO provides a comprehensive pipeline integrating generation, search, and voting in a principled manner

- The paper addresses prompt optimization from multiple angles (generation diversity, search efficiency, robustness via ensembling)

- The focus on black-box optimization makes the approach applicable to commercial LLM APIs

- Consistent improvements across diverse tasks (fact-checking, navigation, hate speech detection, sarcasm detection, reasoning)

- The global tracking of difficult cases across prompts is a useful addition to existing feedback-based methods

**Weaknesses:**

- Lack of statistical testing, incomplete ablations, missing computational cost analysis
- Why should this ensemble approach work? What are the theoretical guarantees? Under what conditions might it fail?
- Embedding methods, optimization procedures, and exact algorithmic integration are underspecified
- The approach requires running multiple generators, search strategies, and ensemble voting. Is the improvement worth this complexity?
- Scalability questions: How does ELPO perform with:
1. Larger candidate pools?
2. More complex tasks requiring longer prompts?
3. Different LLM families (GPT vs Claude vs Llama)?
4. Limited API budgets?

**Questions:**

- Can you provide comprehensive analysis showing search efficiency across all datasets? Specifically: (a) Top-k precision (what % of selected prompts are truly in top-k?), (b) Oracle gap (performance difference between selected prompts and true best), (c) Computational savings in terms of total evaluations?
- Can you provide ablations for: (a) Each generator independently, (b) Each search method independently, (c) Voting strategies (single best vs average vs weighted), (d) Different ensemble sizes?
- Can you report standard deviations, confidence intervals, or conduct significance tests (e.g., paired t-tests) across the 3 runs?
- What embedding model is used for Bayesian Search and MAB? How sensitive are results to this choice?
- What optimization algorithm solves the constrained problem in line 365? How is F1_macro made differentiable w.r.t. weights w?
- What is the total number of LLM API calls required by ELPO vs baselines? What is the wall-clock time and estimated dollar cost?
- How does ELPO perform when: (a) Using different task models (GPT-4, Claude, Llama)? (b) Using different optimizer models? (c) Applied to longer-form generation tasks?
- Can you analyze cases where ELPO performs worse than simpler baselines? What types of tasks or prompts are challenging?

---

### Note · Authors · 2025-11-20

**Comment:**

We sincerely thank the reviewers for their time and feedback. The authors decided to withdraw this submission due to changes in the authors’ submission plan.

**Withdrawal Confirmation:**

I have read and agree with the venue's withdrawal policy on behalf of myself and my co-authors.